# NULISA: a proteomic liquid biopsy platform with attomolar sensitivity and high multiplexing

Wei Feng[1], Joanne C. Beer [1], Qinyu Hao[1], Ishara S. Ariyapala[1], Aparna Sahajan[1], Andrei Komarov[1], Katie Cha[1], Mason Moua[1], Xiaolei Qiu[1], Xiaomei Xu[1], Shweta Iyengar[1], Thu Yoshimura[1], Rajini Nagaraj[1], Li Wang[1], Ming Yu[1], Kate Engel[1], Lucas Zhen[1], Wen Xue[1], Chen-jung Lee[1], Chan Ho Park[1], Cheng Peng[1], Kaiyuan Zhang[1], Adrian Grzybowski[1], Johnnie Hahm[1], Susanne V. Schmidt[2], Alexandru Odainic [2,3], Jasper Spitzer [2], Kasun Buddika[1], Dwight Kuo [1], Lei Fang[1], Bingqing Zhang[1], Steve Chen[1], Eicke Latz [2,4], Yiyuan Yin[1], Yuling Luo [1]✉ & Xiao-Jun Ma [1]✉

The blood proteome holds great promise for precision medicine but poses substantial challenges due to the low abundance of most plasma proteins and the vast dynamic range of the plasma proteome. Here we address these challenges with NUcleic acid Linked Immuno-Sandwich Assay (NULISA™), which improves the sensitivity of traditional proximity ligation assays by ~10,000-fold to attomolar level, by suppressing assay background via a dual capture and release mechanism built into oligonucleotide-conjugated antibodies. Highly multiplexed quantification of both low- and high-abundance proteins spanning a wide dynamic range is achieved by attenuating signals from abundant targets with unconjugated antibodies and next-generation sequencing of barcoded reporter DNA. A 200-plex NULISA containing 124 cytokines and chemokines and other proteins demonstrates superior sensitivity to a proximity extension assay in detecting biologically important low-abundance biomarkers in patients with autoimmune diseases and COVID-19. Fully automated NULISA makes broad and in-depth proteomic analysis easily accessible for research and diagnostic applications.

Blood has been widely used as a source for liquid biopsy, particularly in cancer, where genetic and epigenetic alterations are routinely assessed using circulating cell-free tumor DNA (ctDNA)[1,2]. However, the blood proteome, which contains actively secreted proteins, and the proteomes of other tissues and pathogens[3,4] holds greater promise for providing a real-time snapshot of the functioning of the entire body. Proteins more closely reflect dynamic physiological and pathological processes[5], and blood-based protein biomarkers are broadly applicable for essentially every disease state. However, interrogating the blood proteome is challenging due to the low concentrations (<1 pg/mL) of most proteins and the vast 12-log dynamic range of protein concentration in blood[6]. To date, only ~150 of the estimated >10,000 plasma proteins are in routine diagnostic use[7]. To unlock this vast source of biomarkers, major technological advances in both sensitivity and multiplexing are needed.

Mass spectrometry (MS)[8] and immunoassays are the two pillars of proteomic analysis today but have significant limitations. MS-based

[1]Alamar Biosciences, Inc, Fremont, CA, USA. [2]Institute of Innate Immunity, Medical Faculty, University of Bonn, Bonn, Germany. [3]Department of Microbiology and Immunology, The Peter Doherty Institute for Infection and Immunity, University of Melbourne, Melbourne, Australia. [4]Deutsches Rheuma-Forschungszentrum Berlin (DRFZ), Berlin, Germany. ✉e-mail: yluo@alamarbio.com; xma@alamarbio.com

methods are biased towards high-abundance proteins (>1 ng/mL)[9] and have a narrow dynamic range (4–6 logs). Immunoassays using a sandwich pair of antibodies to recognize a target can achieve higher sensitivity, in the range of 1–10 pg/mL[9] but even the most sensitive of these, capable of single-molecule detection, including immuno-PCR[10], proximity ligation assay (PLA)[11,12], proximity extension assay (PEA)[13], single molecule array (SIMOA)[14], and single molecule counting (SMC)[15], remain inadequate to reach the low abundance portion of the plasma proteome.

Multiplexed immunoassays that simultaneously measure multiple proteins can save both time and precious samples, improve precision, and more importantly, provide a more comprehensive view of the proteome and potentially biological insights. However, high multiplexing poses another considerable challenge due to the exponential increase in cross interactions between noncognate pairs of antibodies[16,17]. Single affinity reagent-based methods, such as the SomaScan assay, can be more easily scaled to high levels of multiplexing but can be prone to false positive signals because of the reliance on a single binding event[18]. Recently, PEA in combination with next-generation sequencing (NGS) was shown to achieve highly specific and multiplexed detection of up to 384 proteins in a single reaction[19]. However, PEA is not currently the most sensitive assay, and the assay protocol consists of numerous manual steps and requires multiple liquid-handling instruments, making it difficult for routine use, especially in clinical settings. Thus, high multiplexing without compromising sensitivity and specificity in a broadly accessible system remains an important unmet need.

Previously, we achieved robust single RNA molecule in situ detection with RNAscope[20], a technology that greatly improves detection sensitivity by suppressing assay background during signal amplification. We hypothesized that the same background suppression principle could be employed in immunoassays to significantly improve protein detection sensitivity.

Here, we report the development of NUcleic acid-Linked Immuno-Sandwich Assay (NULISA), which incorporates multiple mechanisms of background suppression. A multiplex NULISA assay for a panel of 204 proteins, including 124 cytokines, chemokines, and other proteins involved in inflammation and immune response was able to detect previously difficult-to-detect but biologically important, low-abundance biomarkers in patients with autoimmune diseases and COVID-19. The combination of ultra-high sensitivity and high multiplexing in a fully automated platform should democratize in-depth and broad analysis of the blood proteome and lead to biological insights and the identification of biomarkers for the earliest detection of serious diseases.

## Results

### NULISA background suppression enables attomolar level sensitivity

To greatly improve immunoassay sensitivity, we sought to suppress the assay background of PLA[11] by designing oligonucleotide DNA-antibody conjugates such that the DNA elements are used for both immunocomplex purification and reporter DNA generation upon proximity ligation (Fig. 1a). The capture antibody is conjugated with a partially double-stranded DNA containing a poly-A tail and a target-specific molecular identifier (TMI), whereas the detection antibody is conjugated with another partially double-stranded DNA containing a biotin group and a matching TMI (Fig. 1a–1). When both antibodies bind to the target in a sample, an immunocomplex is formed. The immunocomplexes are captured by paramagnetic oligo-dT beads via dT-polyA hybridization (Fig. 1a–2), and the sample matrix and unbound detection antibodies are removed by washing (Fig. 1a–3). As dT-polyA binding is sensitive to salt concentration, the immunocomplexes are then released into a low-salt buffer (Fig. 1a–4). After removing the dT beads, a second set of paramagnetic beads coated with streptavidin is introduced to

capture the immunocomplexes a second time (Fig. 1a–5), allowing subsequent washes to remove unbound capture antibodies, leaving only intact immunocomplexes on the beads. Then, a ligation reaction mix containing T4 DNA ligase and a specific DNA ligator sequence is added to the streptavidin beads, allowing the ligation of the proximal ends of the DNA attached to the paired antibodies and thus generating a DNA reporter molecule containing unique pairs of TMIs (Fig. 1a–6). The DNA reporter can then be quantified by quantitative PCR (qPCR) for single-plex analysis (Fig. 1a–7a) or NGS for multiplex analysis (Fig. 1a–7b). Due to the use of paramagnetic beads, the entire workflow can be readily automated from sample preparation to qPCR data acquisition or to an NGS-ready library.

For NGS analysis of multiple samples, sample-specific molecular identifiers (SMI) are introduced into the reporter DNA using double-stranded ligators containing SMI sequences (Fig. 1a–7b), which enables pooling of a full 96-well plate of samples into one sequencing reaction. After ligation, the ligation products are pooled, amplified, and purified to generate a sequencing library that is compatible with multiple Illumina sequencing instruments. Raw sequencing reads are processed to generate target- and sample-specific counts. These are normalized using spiked-in mCherry protein to control for potential well-to-well variation and inter-plate control samples for plate-to-plate variation.

To evaluate the effectiveness of the dual capture and release mechanism to suppress background in NULISA, we compared NULISA with homogeneous PLA using the same antibody-DNA conjugates and assay buffer to detect IL4 in a serial dilution from 500 pM to 10 aM. Compared to PLA, NULISA reduced the background by more than 10,000-fold, thereby achieving attomolar-level limit of detection (LOD) and a 7-log dynamic range, corresponding to a 4-log increase (Fig. 1b). Similarly, NULISA assays detecting IL6 and CXCL5 demonstrated 22 aM and 26 aM LOD, 30,000- and 3600-fold lower than those from the PLA assays using the same antibody conjugates and reagents, respectively (Supplementary Fig. 1).

We next compared NULISA with a leading single-molecule detection assay, SIMOA, using the same pair of antibodies to detect human immunodeficiency virus (HIV) p24 protein spiked into healthy donor plasma. NULISA demonstrated an LOD of 10 aM (0.24 fg/mL), which is nearly 10-fold lower than that of SIMOA (Fig. 1c); notably, the SIMOA assay used six times more sample than NULISA (124 μL vs. 20 μL). In addition, the standard curve of the NULISA assay demonstrated a dynamic range of >7 logs, 3 logs wider than the range from SIMOA.

To demonstrate the specificity of qPCR-based NULISA, we determined the LOD for human EGFR in the presence of 1 pM to 100 pM mouse Egfr which shares 90% sequence identity with the human protein. The presence of a one-million-fold excess of mouse Egfr did not substantially interfere with the quantification of human EGFR, as the LOD remained well below 100 aM and the assay background showed only a small increase (Supplementary Fig. 2), indicating the robust specificity of NULISA.

These results demonstrated that background suppression by the double immunocomplex purification steps in NULISA enabled attomolar-level sensitivity, and qPCR-based NULISA outperformed the current state-of-the-art in sensitivity and dynamic range. It should be noted that the large dynamic range shown for IL4 and p24 in these experiments was attributable to background suppression and the qPCR readout, but the degree of improvement for other targets will vary depending on the specific target and antibodies used as well as the readout method.

### NULISAseq enables high multiplexing with attomolar sensitivity

We next aimed to demonstrate the performance of multiplexed NULISA with NGS readout (NULISAseq) for detecting a 200-plex inflammation panel that includes mostly cytokines, chemokines, and other inflammation/immunology-related targets, many of which are present at very low levels in the blood[3] (Supplementary Data 1).

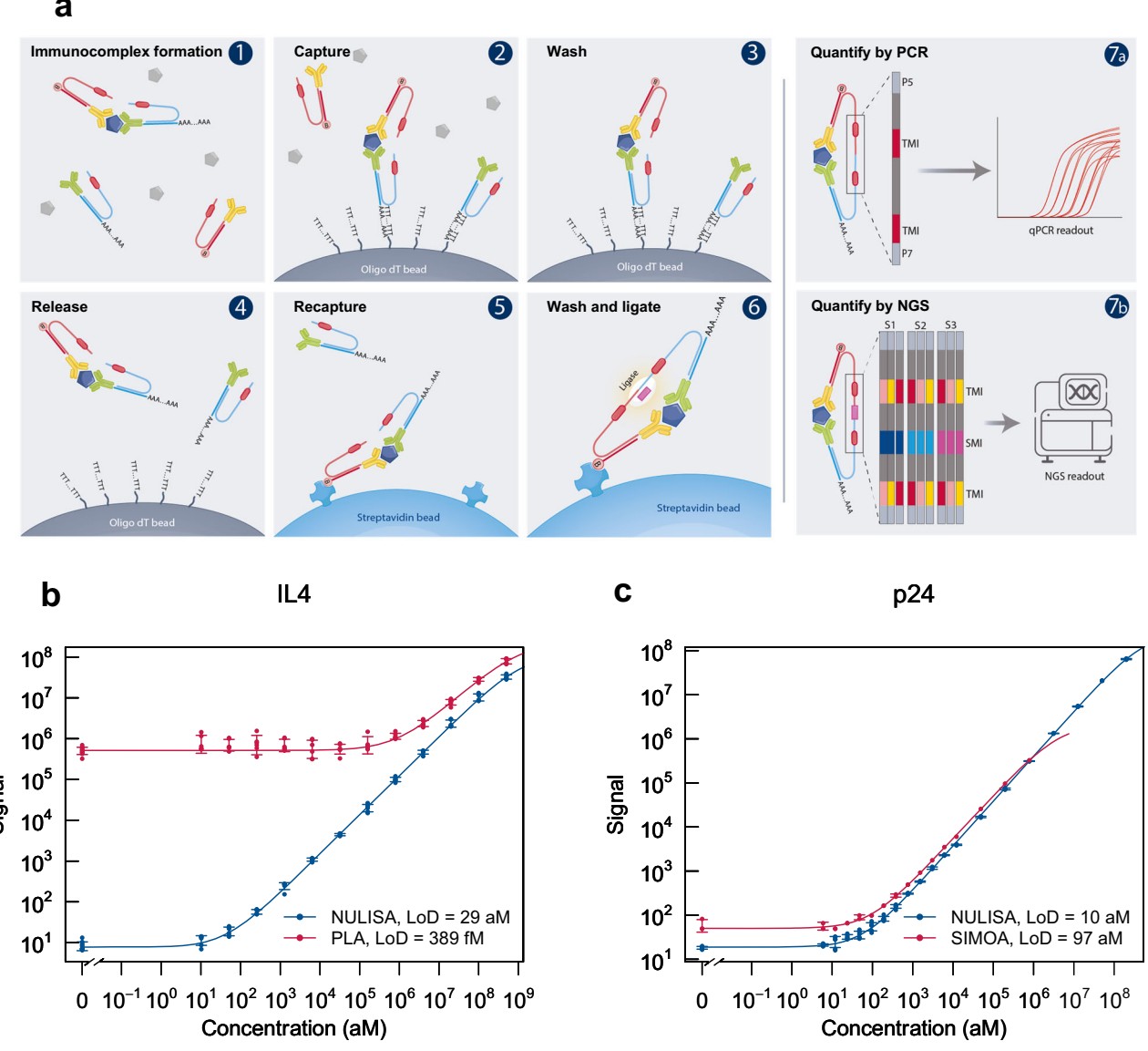

**Fig. 1 | NULISA design and proof of concept. a** Schematic of the NULISA workflow. (1) Immunocomplex formation; (2) first capture of immunocomplexes to dT beads; (3) bead washing to remove unbound antibodies and sample matrix components; (4) release of immunocomplexes into solution; (5) recapture of immunocomplexes onto streptavidin beads; (6) bead washing and DNA strand ligation to generate reporter DNA; (7a) detection and quantification of reporter DNA levels by qPCR; (7b) quantification of reporter DNA levels by NGS. **b** Standard curves for IL4 detection generated following the traditional PLA (red line) or NULISA (blue line) protocols using the same set of reagents. The serial dilution of the standard spanned from 200 pM to 10 aM. Error bars represent mean +/− one standard deviation ($n = 6$). **c** Sensitivity and dynamic range comparison between NULISA (blue line) and SIMOA (red line) using the same pair of antibodies to detect HIV p24. Error bars represent mean +/− one standard deviation ($n = 3$). Source data are provided as a Source data file.

To evaluate the sensitivity of the 200-plex NULISAseq assay, standard curves were generated using a serial dilution of pooled recombinant proteins, and LODs were determined (Supplementary Data 1). For comparison, single-plex standard curves for 11 targets were also constructed using individual recombinant proteins and qPCR-based NULISA with the same antibody reagents. Single-plex and multiplex assays demonstrated similar attomolar-level LODs, e.g., 58 aM and 36 aM for LIF, 20 aM and 3 aM for IL5, and 66 aM and 93 aM for IL13, respectively (Fig. 2a; see Supplementary Fig. 3 for all 11 targets). This indicated that at least for these 11 targets multiplexing was achieved with NULISAseq without a reduction in sensitivity. Furthermore, the data from the single-plex and 200-plex assays demonstrated good correlation in 12 healthy donor plasma samples (Pearson correlation r > 0.8) (Fig. 2b). To determine whether different levels of multiplexing could generate consistent results, we pooled 24

individual assays from the 200-plex assay to create a 24-plex panel and used it to analyze the same 12 samples. The correlations between the results of the 200-plex and 24-plex NULISAseq were also high (r > 0.9) for all targets (Fig. 2c).

These results thus demonstrated that NULISAseq was able to multiplex >200 targets with similar sensitivity to qPCR-based single-plex assays, and the level of multiplexing could be flexibly configured to obtain consistent data.

**Dynamic range, precision, cross-reactivity, and detectability**
The 12-log dynamic range of plasma protein concentrations[3] poses a special challenge for multiplexing with NGS as the readout. Because all targets share the same sequencing library, a few high-abundance targets can consume a large fraction of the total reads, reducing the probability of detecting low-abundance targets with lower reads. To

simultaneously detect both high- and low-abundance targets, it is important to have a relatively even distribution of sequencing reads across all targets. To achieve this goal, we mixed unconjugated "cold" antibodies with DNA-conjugated "hot" antibodies to reduce the

number of sequencing reads from high abundance targets. In a proof-of-concept experiment, the tuned signal from the hot/cold antibody mixture was proportional to the percentage of "hot" antibodies, while the shape of the standard curve was well maintained (Supplementary

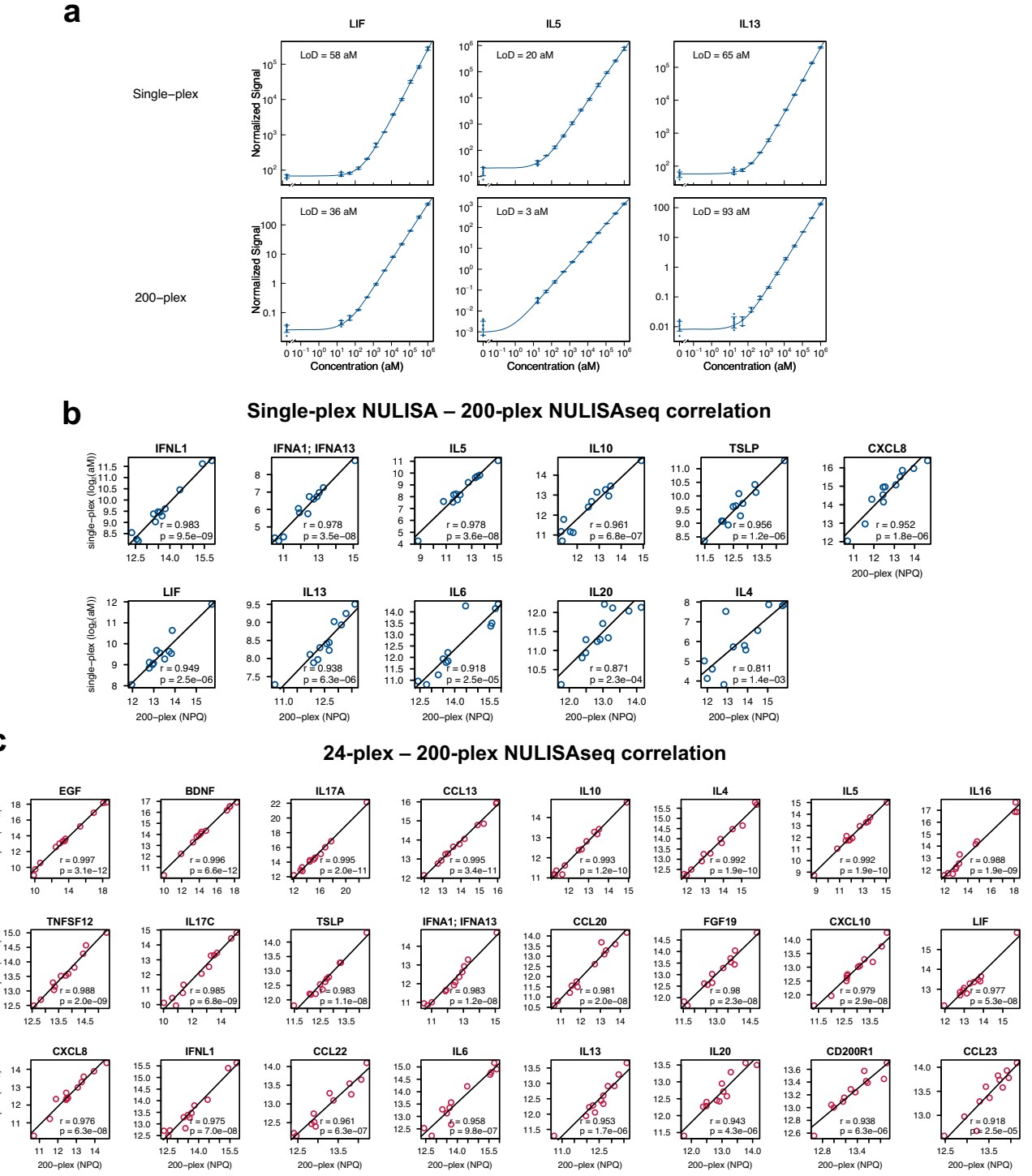

**Fig. 2 | NULISAseq sensitivity and correlation across multiplex levels.**
**a** Comparison of the sensitivity of 200-plex and single-plex assays for LIF, IL5, and IL13 detection. Error bars represent mean +/− one standard deviation ($n = 3$).
**b** Pearson correlation of protein levels measured using 200-plex and single-plex assays with 12 individual healthy donor plasma samples. The 200-plex data were normalized using internal and inter-plate controls and then log2-transformed to yield NULISA Protein Quantification (NPQ) units. Single-plex data in absolute concentration (aM) were also log2-transformed. Least-squares regression lines are shown on the plots. Two-sided tests were carried out to assess whether correlation

coefficients significantly differ from zero; unadjusted *p*-values are shown. **c** Pearson correlation of protein levels measured using 200-plex and 24-plex NULISA methods with the same 12 plasma samples. Data were normalized using internal and inter-plate controls and then log2-transformed to yield NULISA Protein Quantification (NPQ) units. Least-squares regression lines are shown on the plots. Two-sided tests were carried out to assess whether correlation coefficients significantly differ from zero; unadjusted *p*-values are shown. The same antibody concentrations were used in all the assays. Source data are provided as a Source data file.

Fig. 4), indicating that the sensitivity for high-abundance proteins could be tuned down while maintaining quantification. Incorporating this strategy in the NULISAseq 200-plex, we adjusted the hot/cold antibody mixing ratios for different targets according to their endogenous concentrations.

We determined the dynamic range of each assay in NULISAseq from standard curves generated from a serial dilution of pooled recombinant proteins. Most assays showed a dynamic range greater than three logs, covering at least two logs above and one log below the endogenous levels in healthy donors (Fig. 3a, Supplementary Data 1). Note that these dynamic ranges were smaller than the 7 logs shown for single-plex assays (e.g., Fig. 1b, c). This was because in the 200-plex assay the highest concentrations in the standard curves did not reach the true upper limit (ULOQ) for most of the targets due to the need to conserve sequencing capacity, and the lower limits (LLOQ) for those assays tuned with hot/cold mixing were intentionally raised for high abundance targets as explained above. The accumulative dynamic range across all targets spanned 9.6 logs (17 aM or 0.26 fg/mL to 70 nM or 1.63 µg/mL) (Fig. 3a; Supplementary Data 1).

We next determined the precision of the NULISAseq 200-plex assay. The intraplate CV was determined using 9 replicates each of 10 individual healthy donor samples and averaged across seven runs for each target. After internal control normalization, the mean and median intraplate CVs across all targets were 10.2% and 9.2%, respectively (Fig. 3b, Supplementary Data 1). The interplate CV was determined using two runs with the same operator, instrument, and reagent lot, and after intensity normalization, the mean and median interplate CVs were 10.3% and 9.1%, respectively (Fig. 3b, Supplementary Data 1).

To assess potential cross-reactivity of the 200-plex assay, we randomly split 198 targets for which recombinant proteins were available into 45 pools of four or five proteins. We repeated the random assignment to create a second set of 45 pools but required that any two proteins be assigned to the same pool only once between the two sets (Supplementary Fig. 5a). Thus, each protein was present in two of the 90 pools, and it was the only one shared by the two pools. We could thus readily identify a cross-reacting protein from a pool of 4 or 5 proteins by identifying the shared protein between two cross-reacting pools. We also created three additional pools of recombinant proteins not included in the 200-plex assay but homologous to one or more the targeted proteins (Supplementary Fig. 5b–d). Each protein, target or non-target in each pool, was at 20 pM concentration. These 93 pools were analyzed as individual samples with the NULISAseq 200-plex assay. For most assays, only the two pools containing the assigned antigen generated specific signals, whereas other pools generated background-level reads, as expected (Fig. 3c; Supplementary Fig. 5e). We further investigated all identified potential cross-reactions using low- or single-plex assays. Two cross-reactions (IL4 and FGF23) were found to be caused by antigen contamination originating from their vendors. Overall, 91% of the assays showed <1% cross reactivity, defined as cross reactivity = $\frac{\text{maximum signal in nontarget pools} - \text{background}}{\text{mean signal in target pools} - \text{background}} * 100$, where background was the median read count from all non-target pools (Supplementary Fig. 5e; Supplementary Data 1).

A useful measure of assay sensitivity is the detectability of targets in real-world samples, defined as the percentage of samples in which a target signal is detected above the LOD. We performed NULISAseq on 151 samples from 79 healthy donors and 72 patients with different diseases (Sample Sets 1 and 2; Supplementary Tables 1 and 2). The mean detectability across the 204 targets was 93.7%, 94.8%, and 94.4% in healthy, diseased, and combined samples, respectively, and overall, 195 (95.6% [95% CI 91.8–98.0] targets demonstrated >50% detectability (Fig. 3d, Supplementary Data 1).

Taken together, the NULISAseq 200-plex assay covered a 9.6-log accumulative dynamic range of low and high abundance proteins, good intra- and inter-plate precision, low cross-reactivity, and high target detectability in clinical samples.

## NULISAseq comparison with other immunoassays

Having established the analytical performance parameters of NULI-SAseq, we next compared NULISAseq with two commercial multiplex assays, the Olink Explore 384-plex Inflammation Panel and MSD V-PLEX Human Cytokine 44-Plex (which includes five separate 7- to 10-plex assays). Target detectability and inter-platform correlations were determined using plasma samples from 39 healthy donors and 35 patients with different diseases (Sample Set 1 in Supplementary Table 1). NULISAseq demonstrated equal or better detectability than the other two assays for 20 of the 23 targets shared by the three platforms (Fig. 4a). Pairwise correlations of the three platforms were good (NULISA vs. MSD mean $r = 0.75$, median $r = 0.90$; NULISA vs. Olink mean $r = 0.71$, median $r = 0.84$; MSD vs. Olink mean $r = 0.74$, median $r = 0.87$), especially for targets that showed good detectability across all three platforms (Fig. 4a). When detectability differed, the correlation was lower; for example, for IL5, NULISA and MSD both had high detectability (100% and 82.4%, respectively), and the results correlated well ($r = 0.73$, $p < 0.001$), but both results correlated poorly with those obtained from Olink (NULISA $r = 0.08$, $p = 0.50$; MSD $r = 0.10$, $p = 0.45$), which had poor detectability (24.3%). The NULISA CCL3 assay showed poor correlation with the other two platforms because of its low detectability.

We further compared NULISAseq and Olink Explore using 151 samples (Sample Sets 1 and 2 in Supplementary Table 1). Two pooled healthy donor plasma samples with four replicates each were included on each plate to evaluate assay precision. For the 92 common targets between the two panels, the mean and median intraplate CVs were comparable between NULISAseq (mean = 7.3%, sd = 2.4, median = 6.7%) and Olink (mean = 8.4%, sd = 5.9, median = 6.4%), with NULISAseq showing a narrower CV distribution (Fig. 4b). A similar pattern was observed for the interplate CV after intensity normalization (NULISAseq mean = 11.6%, sd = 4.0, median = 11.1%; Olink mean = 12.5%, sd = 7.2, median = 9.9%). The percentage of detectable targets for both panels exceeded 90% (96.7% [95% CI 90.8–99.3] for NULISAseq and 92.4% [95% CI 84.9–96.9] for Olink); however, NULISAseq showed better detectability in samples from both healthy individuals and patients (Supplementary Table 3; Supplementary Fig. 6).

To further compare the sensitivity of NULISAseq and Olink PEA, we compared the NULISAseq LOD/LLOQ data in Supplemental Data 1 with the data in the Olink Explore 3072 validation datasheet (https://olink.com/content/uploads/2023/07/olink-explore-3072-validation-data-results.xlsx). We restricted this comparison to those targets that were shared between NULISAseq 200-plex and Olink Explore panels and excluded those assays for high abundance proteins that were tuned down by hot/cold antibody mixing in NULISAseq or used diluted samples in the PEA assay, resulting in 74 shared targets between the 200-plex and the entire Olink Explore 3072. This comparison thus eliminated the impact of different signal attenuation strategies used by the two platforms and focused on targets requiring high sensitivity. NULISAseq demonstrated significantly lower LODs and LLOQs than PEA overall (Supplementary Fig. 7). The median LOD and LLOQ for PEA were 250-fold and 65-fold higher than that for NULISA, respectively, with large target-dependent variations (Supplementary Data 3). In addition to this in silico comparison, we further examined the subset of 45 of these 74 targets shared between NULISA 200-plex and Olink Explore Inflammation Panel for which we had detectability data on both platforms (Supplementary Data 3). For these 45 targets, the mean detectability was 95% for NULISAseq and 83% for Olink PEA (paired $t$-test $t = 2.5$, degrees of freedom = 44, $p = 0.016$). There was significant overall correlation between LOD ratios and detectability differences (Spearman rho = 0.6, $p = 3.5e-5$). For example, the targets with the largest LOD ratios (Olink to NULISA ratio), IL4, IL5, IL13, IL20 and IL33, also had the largest differences in detectability (Supplementary Data 3). These targets all have low or sub pg/mL concentrations in blood according to Human Protein Atlas (https://proteinatlas.org).

Taken together, NULISAseq demonstrated higher sensitivity and higher detectability than Olink PEA.

We next compared the ability of NULISAseq and Olink PEA to identify changes in protein abundance between patients with inflammatory diseases ($n = 21$) and healthy controls ($n = 79$). When all targets in the NULISAseq 200-plex and Olink 384-plex inflammation panels were evaluated in linear models, 56% (114 out of 204) of the NULISAseq targets and 26% (94 out of 368) of the Olink targets

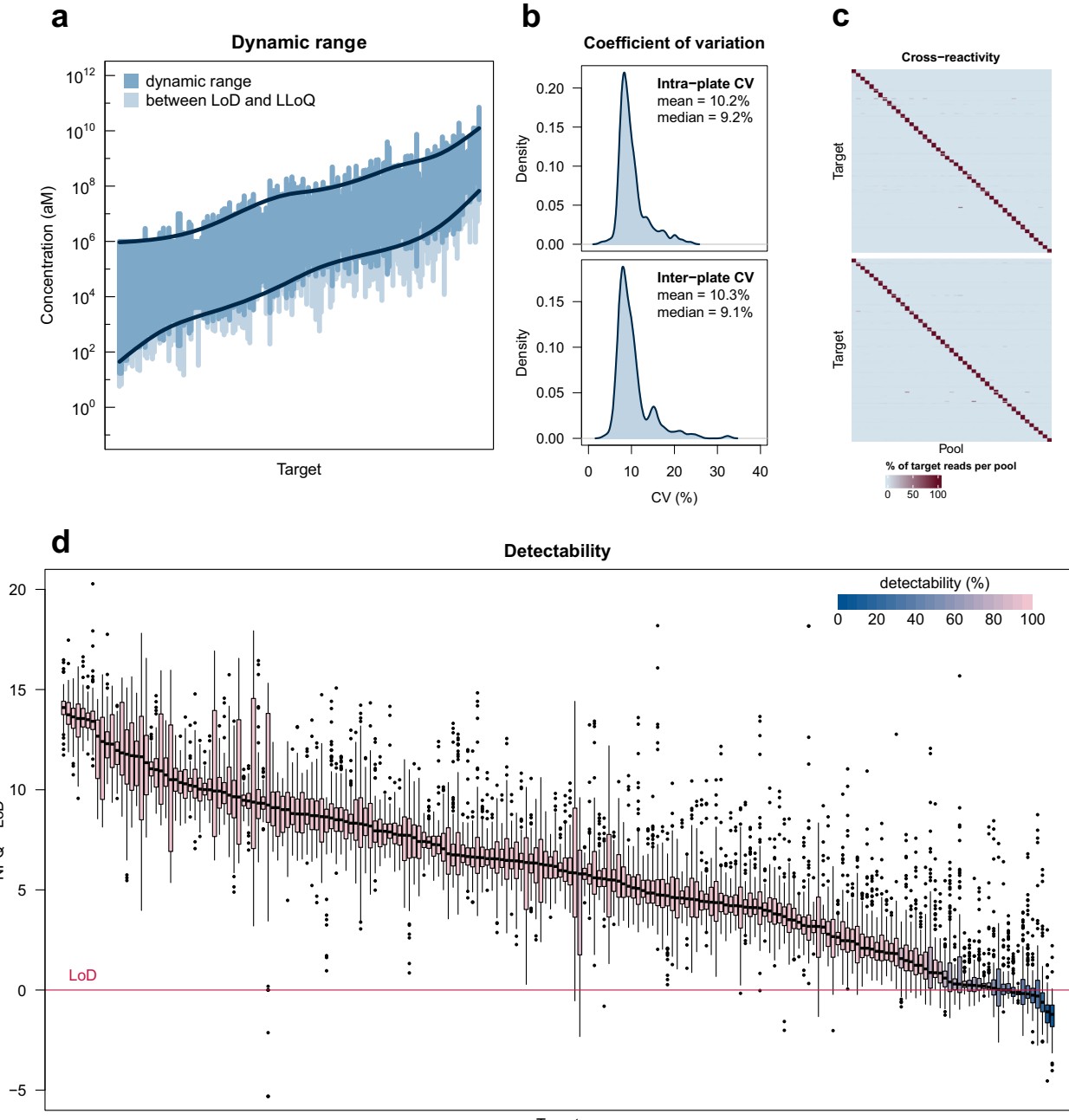

**Fig. 3 | NULISAseq performance characterization. a** The dynamic range for the detection of each target is indicated by the dark blue region. Values above the limit of detection (LoD) but below the lower limit of quantitation (LLoQ) are shown in lighter blue. Targets are ordered according to the geometric mean of the LLoQ and upper limit of quantitation (ULoQ). Dark blue lines represent generalized additive model (GAM) cubic regression spline fits to the ULoQ and LLoQ. The overall dynamic range for 200-plex NULISAseq spanned 9.6 log10 values. **b** Density plots of intraplate CV after internal control normalization and interplate CV after internal control and intensity normalization. CV for each target was calculated using the mean CV across 10 samples with 9 technical replicates each. **c** Cross reactivity. Two sets of 45 random antigen pools containing 4–5 targets each were analyzed with 200-plex NULISAseq. Each cell of the heatmap represents the percent of normalized read counts for that target (rows) occurring in that pool (columns) (each row adds up to 100); scale ranges from zero (light blue) to 100 (dark red). Targets were ordered according to pool membership such that the cells on the diagonal corresponded to the assigned pools. **d** Detectability of 204 targets in 151 samples, including 79 from healthy controls and 72 from patients with various diseases. The *y*-axis represents NULISA Protein Quantification (NPQ) units minus the LoD for the respective target. Boxplots show lines at median; boxes indicate interquartile range; whiskers show values extending from interquartile range to up to 1.5 times the interquartile range; data beyond this are shown as plotted points. Boxplots are shaded using a gradient scale corresponding to detectability, where light pink represents 100% detectability and dark blue represents 0% detectability. Source data are provided as a Source data file. Source data for (**a**, **b**) are in Supplementary Data 1. Source data for (**d**) are provided in the Alamar_NULISAseq_Detectability_NPQ.csv file available under accession GSM7734324.

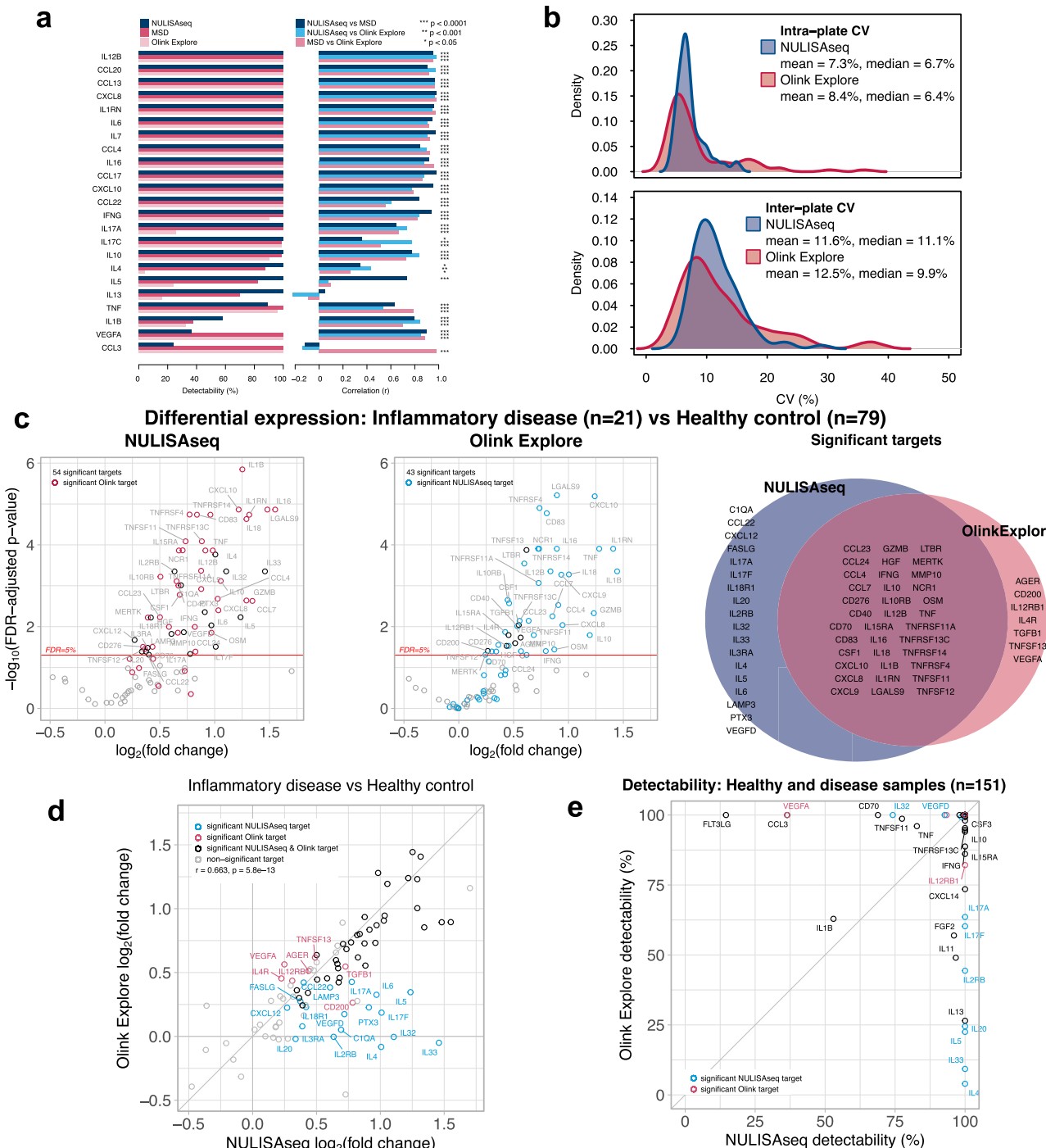

**Fig. 4 | Comparison of NULISAseq with other assay platforms. a** Detectability and inter-platform correlation for 23 shared targets in the NULISAseq 200-plex, Olink Explore 384 Inflammation Panel, and MSD V-PLEX Human Cytokine 44-Plex assays. Two-sided tests were carried out to assess whether correlation coefficients significantly differ from zero; unadjusted $p$-values are shown. Exact $p$-values are listed in Source data file figure_4a_summary_data.csv. **b** Intra- and interplate CV distributions for the 92 common targets in the NULISAseq 200-plex (shown in blue) and Olink Explore 384-plex (shown in red). CV for each target was calculated as the mean CV for two independent pooled plasma samples with four technical replicates each. **c** Volcano plots of -log10(FDR-adjusted $p$-value) versus log2(fold change) levels comparing protein abundances in samples from patients with inflammatory diseases ($n = 21$) and healthy controls ($n = 79$) with NULISAseq and Olink Explore. Black open circles represent targets that were uniquely significant for the specified panel; targets that were significant in the other panel are shown as red (significant

Olink target) or blue (significant NULISAseq target) open circles. The Venn diagram shows the overlap of significant targets. Two-sided significance tests were carried out to assess whether log2(fold change) differed from zero for each target; $p$-values were adjusted using a false discovery rate correction. **d** Correlation of estimated log2-fold changes between NULISAseq and Olink Explore. Targets are highlighted in blue (detected by NULISAseq only), red (detected by Olink only), black (detected by both panels), or gray (not significant for either panel). A two-sided test was carried out to assess whether the Pearson correlation coefficient significantly differs from zero. **e** Comparison of detectability between NULISAseq and Olink Explore, assessed using the same 79 healthy control samples and 72 samples from patients with inflammatory and other diseases. Targets are highlighted in blue (detected by NULISAseq only) or red (detected by Olink only); other targets are in black. Source data are provided as a Source data file. Source data for (**b**–**e**) are in Supplementary Data 2.

demonstrated change in protein abundance at a 5% false discovery rate (FDR) significance threshold, respectively (Supplementary Fig. 8). For the 92 common targets, NULISAseq and Olink identified 54 and 43 significantly different targets, respectively, 36 of which were identified using both platforms (Fig. 4c). The correlation of the estimated fold-changes by the two platforms was also significant ($r = 0.67$, $p < 0.001$) (Fig. 4d). The largest differences were observed for some of the targets that were poorly detected by the Olink assay (IL4, IL5, IL20, IL17A, IL17F, IL33, and IL2RB) (Fig. 4e). For NULISAseq, IL4 levels were elevated in all four groups of autoimmune diseases (rheumatoid arthritis, Sjögren's syndrome, systemic lupus erythematosus and ulcerative colitis) (Supplemental Fig. 9). IL4, IL17 and IL33 all have well established roles in autoimmune diseases[21–23]. These results thus indicate that the higher sensitivity of NULISAseq can identify more clinically important but low-abundance biomarkers.

## Characterization of the host response to SARS-CoV-2 with NULISAseq

With the improved sensitivity of NULISAseq and the broad coverage of cytokines and chemokines and their receptors in the 200-plex panel, we next sought to characterize the host response to SARS-CoV-2 infection in patients with a mild COVID-19 disease course. We analyzed 46 serum samples from 9 patients with mild COVID-19 and 16 control samples from healthy donors from a cohort previously studied for the development of anti-spike antibodies[24] (Supplementary Table 4).

Protein abundance at each time interval (relative to the time of peak SARS-CoV-2 nucleocapsid protein expression) for COVID-19 patients was compared with controls using a linear mixed model adjusted for age and sex and accounting for repeated measures[25]. This analysis identified a total of 88 significantly different targets (FDR-adjusted $p < 0.05$ per time interval; 86 with elevated and two with decreased levels in COVID-19 patients; Supplementary Data 3). A group of interferons, including type I interferons (IFNA1, IFNA2, IFNW1) and type III interferon IFNL1, exhibited large (11 to 500-fold) and coordinated increases in abundance across the first three time points (indicated by * in Fig. 5a, b), indicating a robust and sustained interferon response characteristic for anti-viral immune responses[26]. Two other interferons, IFNB1 (type I) and IFNG (type II), were also among the top 20 targets with the biggest changes in protein abundance (Fig. 5b).

The top 20 significant targets also included CXCL10 (IP-10), IL6 and IL10, a cytokine-chemokine triad associated with severe COVID-19[27], as well as C1QA, an important component of the complement system previously shown to be present at elevated levels in SARS-CoV-2-infected patients[28]. The 88 differential targets also included 11 (CCL16, CCL7, CXCL10, CCL8, IL1RN, CD274, IL6, IL18, MERTK, IFNG, and IL18R1, indicated with ** in Fig. 5a) of the 14 proteins detected in COVID-19 patients in multiple previous studies[29], 13 of which were included in the 200-plex panel, although most of the patients in prior studies had more severe disease[30]. Gene ontology (GO)[31] analysis of the 86 targets with elevated levels indicated enrichment in leukocyte proliferation (GO:0070661), cell activation involved in immune response (GO:0002263), and humoral immune response (GO:0006959) (Supplementary Data 4), indicating the activation of both innate and adaptive immune response mechanisms in response to SARS-CoV-2 infection.

These results indicated that NULISAseq was able to detect important low abundance proteins such as interferons and recapitulate many key aspects of the host response to SARS-CoV-2.

## Discussion

Our results demonstrate that NULISA achieves attomolar-level sensitivity and can measure hundreds of proteins simultaneously using 10-20 µL sample in 96-well plate format. NULISA's ultra-high sensitivity is largely driven by an ~10,000-fold reduction in assay background compared to traditional PLA, which is enabled by the molecular design

of the DNA-antibody conjugate that provides a mechanism to purify the immunocomplexes prior to proximity ligation.

Several design features of NULISA simplify multiplexing and automation. NULISA generates the reporter DNA molecule by ligation, which makes it easy to design highly multiplexed assays. We demonstrated multiplexed detection of >200 targets, and we have designed >6000 target-specific barcodes for much higher levels of multiplexing in the future. For comparison, PEA[32] requires hybridization between the two oligonucleotides linked to the antibody pair to initiate primer extension, and the length of the hybridized sequence needs to be short to prevent antigen-independent cross-hybridization (e.g., 5 base pairs provide $4^5 = 1024$ possible sequences) yet unique enough to provide hybridization specificity for each antibody pair, further limiting the number of available sequences for multiplexing. The use of paramagnetic beads in NULISA also makes the protocol more amenable to automation by standard liquid handling equipment from commercial vendors, such as the TECAN and Hamilton instruments; for example, this study was performed using an in-house prototype system based on a Hamilton instrument. The automated protocol has a 6-h run time and <30 min hands-on time from sample preparation to the acquisition of qPCR results or the preparation of a PCR-amplified DNA library ready for NGS sequencing. In comparison, Olink Explore is semi-automated requiring multiple different instruments and >5-h hands-on time over two days.

The large dynamic range of plasma protein concentrations poses another challenge to multiplexing high and low abundance proteins in a single reaction, especially when using NGS as a readout. To prevent high abundance targets from dominating the sequence library, we use a signal tuning strategy to equalize sequence reads across different targets. Using this approach, the NULISAseq 200-plex assay demonstrated an accumulative 9.6-log dynamic range across all targets without sample dilution. This approach greatly simplifies the creation of flexible multiplex panels from a large library of assays, which should facilitate translational studies from discovery to validation.

As the first application of this platform, we developed a 200-plex inflammation and immune response-focused panel including 124 cytokines and chemokines and other important inflammatory and immune-related proteins. To the best of our knowledge, this panel provides the broadest coverage of cytokines and chemokines by any single inflammation-focused panel. For example, it includes all three interferon types with high sample detectability as demonstrated in our COVID-19 study: IFNA1 (100%), IFNA2 (69%), IFNB1 (100%), IFNW1 (78%), IFNG (100%), and IFNL1 (100%). Type I and type III interferons are the body's first-line defense against viral replication and spreading at the site of infection[33,34], but their detection in the blood requires ultra-high sensitivity[35]. A study[30] using Olink Explore included only IFNG and IFNL1, but the detection rates were lower at 76% for IFNG and 19% for IFNL1. One previous study was unable to detect IFNB and IFNG in COVID-19 patient sera using traditional ELISA[36], and another study using a more sensitive SIMOA-based assay was able to detect IFNA in some patients but not in controls[27]. To our knowledge, the current study is the first to report the detection of IFNW1 in circulation and the detection of all three interferon types in a single assay, demonstrating a robust interferon response in mild COVID-19 patients. In fact, the different assays with different sensitivities used in previous studies might have contributed to discrepant findings regarding the relationship between interferons and disease severity[35]. Some studies found a less pronounced interferon alpha response in patients with severe disease than in those with mild disease[36,37], another study reported persistently higher interferon alpha and lambda levels in patients with severe disease than in those with moderate disease[38], and yet another study found no association between interferon alpha levels and disease severity[39]. These findings highlight the critical importance of highly sensitive and robust assays in exploring the low-abundance

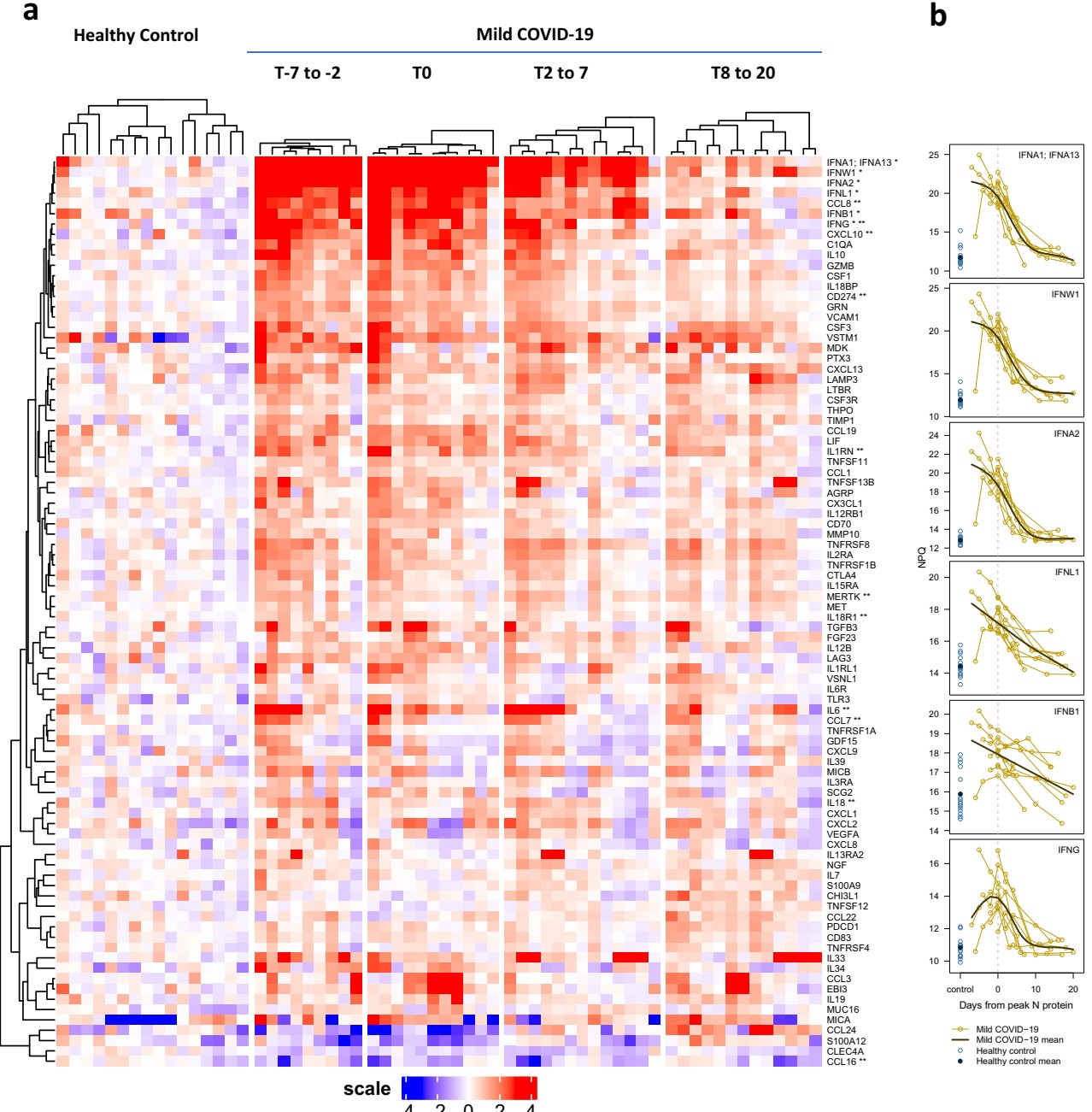

**Fig. 5 | Heatmap of differential protein abundance between SARS-CoV-2-infected patients and healthy controls at different time points.** T0 (*n* = 9) represents time of peak expression of SARS-CoV-2 nucleocapsid protein (N-protein); T-7 to -2 (*n* = 11) represents 2–7 days before T0, T2 to 7 represents 2–7 days after T0 (*n* = 13), and T8 to 20 represents 8–20 days after T0 (*n* = 13). Mixed effect linear model analysis was performed for each time point comparing COVID-19 samples to healthy controls (*n* = 16). **a** A total of 88 differential abundance proteins were identified and visualized in a clustered heatmap. The heatmap displays the log2-normalized read counts (NULISA Protein Quantification, NPQ) centered relative to the mean in the healthy control samples; values above the mean are shown in red and values below the mean are shown in blue. Target names provided in the text are marked with asterisks: * indicates interferons, and ** indicates targets previously reported to be associated with COVID-19. **b** Individual trajectories of interferon abundance (NPQ) relative to days from peak N-protein are shown for the mild COVID group (gold lines). Trajectories are aligned based on the day of peak N-protein (dashed gray vertical line). Generalized additive models with a cubic regression spline basis were used to estimate the mean mild COVID-19 trajectory (dark brown line). Control samples (lighter blue open circles) and control group mean (dark blue solid circle) are shown to the left. Source data are provided in the Alamar_NULISAseq_COVID_NPQ.csv file available under accession GSM7734324.

portion of the proteome that is enriched for important cytokines, such as interferons[40].

As indicated by the low sensitivity of the NULISA CCL3 assay used in this study, antibody affinity and specificity still play an important role. Switching to a different antibody pair improved the detectability of this assay to 100% (Supplementary Fig. 10). In our cross-platform comparison analyses, e.g., comparing NULISA and PEA, the different antibodies used in different platforms, in addition to assay design differences, could also contributed to differences in sensitivity in favor of one platform or another. Further efforts are needed to generate antibodies with high affinity and specificity in a predictable and scalable manner, which may be necessary for the analysis of low-

abundance biomarkers in demanding applications, such as early disease detection.

In conclusion, NULISA represents a significant advance in sensitivity, multiplexing and simple to use automation in a proteomic platform. With a focus on low abundance proteins in blood, such as the cytokine/chemokine-rich 200-plex inflammation panel, NULISA has potential for both biomarker discovery and validation for liquid biopsy-based applications from early detection of diseases to informing prognosis, therapy selection and patient outcome, and even leading to a deeper understanding of human biology at the system level.

# Methods

## Ethical statement

Blood sampling of SARS-CoV-2-infected patients and healthy controls was approved by the local institutional research ethics board (University Hospital Bonn, ethics vote 468/20). We collected comprehensive clinical and demographic information, medical history, comorbidity information, and vaccination schedules for all patients and participants. All participants provided informed consent to participate. This study was conducted in accordance with the principles of the Declaration of Helsinki.

## Patients and samples

Human EDTA plasma samples from healthy donors and patients with various diseases (Supplementary Table 1) were purchased from BioIVT (Westbury, NY) and stored at -80 °C. Patient demographics and clinical diagnostic information (Supplementary Table 2) were provided by BioIVT.

The patients in the COVID-19 study were a subset of the mild disease cohort described previously[24]. In brief, samples were collected from January to March 2020 up to six times during the 21 days from inclusion in a study in Germany in the region of North Rhine-Westphalia. SARS-CoV-2 infection status was confirmed in all patients by qRT–PCR or antigen tests for SARS-CoV-2 using nasopharyngeal swabs and quantification of SARS-CoV-2 nucleocapsid (N) protein concentrations and serum anti-N Ab titers.

## Antibody-oligonucleotide conjugation and signal tuning

Most of the antibodies (>90%) in the 200-plex assay were rabbit or mouse monoclonal antibodies, and the remaining were goat or rabbit polyclonal antibodies (Supplementary Data 5).

Antibody-oligonucleotide conjugation was performed based on amine-to-sulfhydryl cross-linking chemistry. For each target, capture and detection antibodies were conjugated to partially double-stranded DNA containing a polyA-containing oligonucleotide and a biotin-modified oligonucleotide, respectively. Antibody-oligonucleotide conjugates were purified to remove unreacted antibodies and oligonucleotides. The conjugates were assessed on protein and DNA gels to ensure >90% purity and successful conjugation. To ensure consistency of this process and provide higher throughput, the conjugation and purification processes were fully automated, and each batch included IL4 antibodies as run controls.

For assays requiring signal down tuning in the NULISAseq 200-plex inflammation panel, the "hot" DNA-antibody conjugate was mixed with the corresponding nonconjugated "cold" antibodies at a ratio that was tuned to cover the range from -10-fold below to -100-fold above the baseline level in healthy donor plasma. The capture and detection antibodies for all targets in the 200-plex assay were pooled.

## NULISA and NULISAseq assay procedures

NULISA reagents, including Assay Diluent (AD), Wash Buffer (WB), Elution Buffer 1 (EB1), oligo-dT paramagnetic beads (dT beads), streptavidin-coated paramagnetic beads (streptavidin beads), Elution Buffer 2 (EB2), Ligation Master Mix (LMM), were developed or commercially sourced by Alamar Biosciences. Plasma samples were briefly centrifuged at $10,000 \times g$ for 10 min or filtered through a 1.2-μm filter plate (Millipore Sigma, Cat. No. MSBVN1210) prior to use. For single-plex NULISA, 10 or 20 μL of sample was added to a reaction mixture (total volume = 100 μL) containing capture and detection antibody conjugates and AD buffer that included heterophilic blockers. For NULISAseq, 10 μL of sample was used in a 100 μL reaction containing capture and detection antibody cocktails in AD buffer. The reaction mixture was incubated at room temperature for 1 h to allow the formation of immunocomplexes.

After immunocomplex formation, the following NULISA workflow was carried out automatically by a preprogrammed MicroLab STARlet liquid handler (Hamilton Company, Reno. NV), which was integrated with a KingFisher Presto magnetic bead processor (Thermo Fisher, Waltham, MA) and a BioTek ELx405 microplate washer (Agilent, Santa Clara, CA). In brief, 10 μL of 10x dT beads was added to the reaction and incubated at room temperature for 1 h, allowing capture of the immunocomplex on the bead surface. The beads were then collected by KingFisher Presto magnetic head and washed three times with WB. Immunocomplex-bound dT beads were then transferred into 65 μL EB1 buffer and incubated for 10 min to release the immunocomplex from the bead surface. The dT beads were then removed and the eluate was then incubated with 10 μL of 10x streptavidin beads for 10 min at room temperature to recapture the immunocomplex using the biotin-containing detection antibody conjugate. After additional washing with WB, the SA beads were incubated with the ligation reagent LMM and a ligator sequence in 50 uL at room temperature for 10 min to generate the ligated reporter oligonucleotide. For single-plex NULISA the same ligator sequence was used for all samples. For NULISAseq, a ligator sequence containing a unique sample barcode was used for each sample analyzed. After ligation and one additional wash, reporter molecules were eluted from the SA beads using 50 μL EB2 buffer for 10 min at room temperature. The final eluate was collected for quantification by qPCR for single-plex NULISA and by NGS for NULISAseq.

For qPCR analysis, 10 μL of the final eluted sample was added to a total of 30 μL containing PCR Master Mix (PMM, Alamar Biosciences). PCR samples were loaded onto a CFX96 Real-Time PCR instrument (Bio-Rad Laboratories, Hercules, CA, USA). The following PCR conditions were used: 30 °C for 5 min; 95 °C for 2 min; and 45 cycles of 95 °C for 3 s and 65 °C for 30 s. The Cq values for each sample were acquired using the Bio-Rad Maestro software.

For NGS analysis, a library was prepared by pooling the reporter molecules from each NULISAseq reaction and then amplifying the products by 16 cycles of PCR. The library was cleaned using Ampure XP reagent (Beckman Coulter, Indianapolis, IN), following the manufacturer's protocol. The library was then quantified using the Qubit 1X dsDNA HS assay kit (Thermo Fisher, Waltham, MA) before being loaded on a NextSeq 1000/2000 instrument (Illumina, San Diego, CA). For the NULISAseq 200-plex inflammation panel, a P2 reagent kit (100 cycles) (Illumina) was used for NGS.

## Proximity ligation assay

The PLA reaction was set up in the same way as for single-plex NULISA. The immunocomplex formation step was performed by incubating the reaction at room temperature for 2 h, which is the same as the combined time for immunocomplex formation and dT bead capture steps in NULISA. After incubation, the reaction was diluted 100 times before the ligation step was performed under the same conditions as those used for NULISA. Ligation was terminated by heating at 75 °C for 15 min. Then, 10 μL of the final sample was used for qPCR analysis.

## Cross-platform comparison

The Simoa® HIV p24 assays were performed by Frontage Laboratories (Exton, PA) using the Simoa® HIV p24 Advantage kit (Quanterix, Billerica, MA). To compare the performance of the HIV p24 assay, a panel

of 16 samples was created by spiking various levels of recombinant p24 into pooled normal plasma, which was then tested using both SIMOA (Frontage) and NULISA (Alamar Biosciences) platforms.

Olink PEA was performed by Fulgent Genetics (Temple City, CA, USA) and the High-Throughput Biomarker Core of Vanderbilt University Medical Center (Nashville, TN, USA), both of which are designated Olink service providers. Seventy-two samples from patients with diagnosed diseases were grouped into broad diagnostic categories (Supplementary Table 1). Stratified sampling was used to randomize the assignment of 79 samples from healthy volunteers and 72 samples from patients with different diseases to each of the two plates (Supplementary Table 2). The same set of samples was profiled using the NULISAseq 200-plex Inflammation panel and the Olink Explore 384 Inflammation kit (Olink Proteomics, Uppsala, Sweden).

The V-PLEX Human Cytokine 44-Plex assay (Meso Scale Discovery (MSD), Rockville, MD, USA) was performed by DC3 Therapeutics (South San Francisco, CA, USA), a qualified MSD service provider, using five V-PLEX kits (Chemokine Panel 1, Cytokine Panel 1, Cytokine Panel 2, Proinflammatory Panel 1, and TH17 Panel 1). A subset of 39 samples from healthy volunteers and 35 samples from patients with different diseases from the above NULISAseq vs. Olink comparison study (Supplementary Table 1, Sample Set 1) was used to compare detectability and assess correlation for 23 common targets across the three platforms.

### NULISAseq data processing and normalization

NGS data were processed using the NULISAseq algorithm (Alamar Biosciences). The sample- (SMI) and target-specific (TMI) barcodes were quantified, and up to two mismatching bases or one indel and one mismatch were allowed. Intraplate normalization was performed by dividing the target counts + 1 for each sample well by that well's internal control (mCherry) counts + 1 (i.e., Normalized Signal). Interplate normalization was performed using interplate control (IPC) normalization or intensity normalization (IN). For IPC normalization, counts were divided by target-specific medians of the three IPC wells on that plate, and then rescaled by the factor $10^4$. To facilitate statistical analyses, IPC-normalized counts were log2-transformed to form a more normal distribution. These log2 IPC-normalized counts are referred to as NULISA Protein Quantification (NPQ) units. For IN, counts were divided by target-specific medians of all samples on a plate and then multiplied by global target-specific medians across all plates. IN counts were also log2 transformed for statistical analyses.

### NULISA and NULISAseq standard curve and LOD determination

For single-plex NULISA, Cq values were transformed to "Normalized Signal"=$2^{(37-Cq)}$ prior to 4PL curve fitting with $1/y^2$ error weighting. The LOD was calculated as 2.5 times the standard deviation of the blank samples (to facilitate direct comparison with Quanterix SIMOA) plus either the mean of the blanks or the y-intercept of the curve fit. These values were backfitted, and the maximum value was used to define the LOD in aM. To determine the LOD of NULISAseq in aM, Normalized Signal was fitted with a 4PL curve-fitting algorithm. This LOD was calculated as described above but used three times the standard deviation of the blanks (for comparison with MSD V-PLEX and Olink Explore).

### 200-plex, 24-plex, and single-plex NULISA correlation

Pearson correlation was calculated for each common target using the mean of two replicates each from 12 healthy donor samples. To ensure more normally distributed data, correlation was assessed using log2-transformed absolute concentrations for single-plex and NPQ for NULISAseq.

### NULISAseq 200-plex dynamic range

Dynamic range was determined for 197 NULISAseq targets using 12-point standard curves. The dynamic range interval was defined as the largest interval in which the coefficient of variation (CV) of all standards fell below 30% and the recovery was within 30%. A maximum of one standard, except for the lower and upper limit of quantitation (LLOQ and ULOQ), was allowed to fall outside the allowable recovery limits. LOD for each target in aM was determined as described above.

### NULISAseq 200-plex precision

Intraplate CV was calculated using (unlogged) normalized counts for each of the 204 targets using nine replicates each of 10 healthy donor samples. The run was repeated seven times, and the target-specific mean intraplate CVs were calculated. Interplate CVs were calculated after intensity normalization for two of these runs that used the same operator, instrument, antibody lot, and reagent lot. Values below LOD were excluded from CV calculations.

### NULISAseq 200-plex cross-reactivity

To assess cross-reactivity, 198 targets were randomly assigned to two sets of 45 pools containing either four or five targets each, so that no two targets shared a pool for both sets. Three additional antigen pools were created to assess cross reactivity with homologous proteins not targeted by the 200-plex inflammation panel (Supplementary Fig. 5). Counts were normalized using the internal control. Cross-reactivity for each target was quantified as (maximum non-target pool count – background)/(average target pool counts – background) * 100, where the background was calculated as the median count across non-target pools.

### NULISAseq 200-plex detectability

To calculate detectability, plate-specific LODs were determined for each of the 204 NULISAseq targets. Detectability for each target was calculated as the percentage of samples above LOD. A target was considered detectable when more than 50% of samples were above LOD. The same procedure was followed for Olink Explore and MSD V-Plex Human Cytokine 44-Plex data using LODs provided in the data output. The detectability of the 23 common targets between the NULISAseq, Olink Explore 384 and MSD V-Plex Human Cytokine 44-Plex was calculated, and Pearson correlation was assessed on the log2 scale.

### Precision comparison of NULISAseq versus Olink PEA

For NULISAseq data, intraplate CVs were calculated based on normalized data for each target using two pooled plasma samples (four replicates each). Then target-wise mean CVs across the two samples were calculated. The same procedure was followed for Olink data, but since Olink Explore NPX data are on the $\log_2$ scale, intraplate CVs were calculated using the formula $CV_j = 100\sqrt{\exp(\sigma_j^2) - 1}$, where $\sigma_j^2 = sd(target_j\,NPX) \cdot \ln2$ as described[19]. Interplate CVs were calculated in similar fashions, respectively, for NULISAseq and Olink NPX data after applying intensity normalization. Values below LOD were excluded from CV calculations for both platforms.

### Sensitivity comparison of NULISAseq versus Olink PEA

To compare LODs and LLOQs between NULISAseq 200-plex and Olink Explore 3072, NULISAseq LODs and LLOQs were from Supplemental Data 1, and Olink LODs and LLOQs were from Olink's Explore 3072 validation datasheet (https://olink.com/content/uploads/2023/07/olink-explore-3072-validation-data-results.xlsx). Excluding assays that were tuned down with hot and cold mixing in NULISAseq and those requiring sample dilution in Olink Explore, 74 shared targets between the two platforms were identified (Supplementary Data 3). NULISA and Olink LOD and LLOQ data for these targets were compared using the nonparametric paired Wilcoxon test. Limiting the shared targets to those between NULISAseq 200-plex and Olink Explore Inflammation Panel resulted in 45 targets.

## Differential protein abundance in inflammatory diseases

Differential protein abundance for inflammatory disease patients relative to healthy controls was assessed using NPQ and intensity-normalized NPX data for NULISAseq and Olink, respectively. For each of the 92 shared targets, a linear regression model was fit to measure the impact of disease status, adjusted for age, sex, and plate. Benjamini-Hochberg FDR correction[41] was applied and statistical significance was defined using a 5% FDR cutoff.

## Characterization of the host response to SARS-CoV-2 infection with NULISAseq

Due to uncertainty of the exact onset of infection, timepoints were synchronized across individuals by setting the timepoint with peak expression of SARS-CoV-2 nucleocapsid protein (N protein) as $t_0$, as described previously[24]. Two cases with no detectable N protein at all timepoints were excluded, so nine mild COVID cases were included in further analyses. Time points 2–7 days before $t_0$ were classified as $t_{-1}$, 2–7 days after $t_0$ as $t_1$, and 8–20 days after $t_0$ as $t_2$. Differential protein expression for mild COVID samples from each time interval compared to 16 healthy donor samples was assessed using mixed-effect linear models with age and sex as covariates and subject as a random effect using the limma R package[25]. Statistical significance was defined as an FDR-adjusted $p$-value < 0.05. Clustered heatmap visualization was performed using the ComplexHeatmap R package[42]. Both target and sample clustering were performed using cosine correlation distance and complete linkage. Differentially expressed proteins were analyzed for pathway enrichment using the Metascape web tool (https://metascape.org)[43]. The multiple gene list option was used to set the full 204 panel targets as background.

## Statistics and reproducibility

No statistical method was used to predetermine sample size. Patient samples were randomized on the assay plate and run as singlets in NULISA and Olink runs. A sample control (pooled plasma) was run in duplicate on each plate to assess precision and reproducibility. The investigators were not blinded to allocation during experiments and outcome assessment. All statistical analyses were performed using R software (v.4.2.0, R Core Team, 2021)[44]. All statistical tests were two-sided.

## Reporting summary

Further information on research design is available in the Nature Portfolio Reporting Summary linked to this article.

## Data availability

All data generated and analyzed in this study are included in this manuscript, its supplementary information, or have been made available in public repositories. Reference concentrations of blood proteins are available at Human Protein Atlas (https://www.proteinatlas.org/humanproteome/blood+protein). The NULISAseq 200-plex Inflammation Panel data and Olink Explore Inflammation Panel data generated in this study have been deposited in the National Center for Biotechnology Information (NCBI) Gene Expression Omnibus (GEO) database under accession code GSE241717. Unless otherwise noted, all figure source data are provided in source_data.zip. Source data are provided with this paper.

## Code availability

BioRad CFX Maestro Manager 2.2 (v4.2.008.0222) was used for qPCR data collection. Illumina BaseSpace Sequence Hub (v7.9.0) and ARGO Command Center (v0.2.0.0) were used for NGS data collection. NULISAseq data was analyzed with the open-source R package NULISAseqR (v1.0)[45] (GPLv3) available at https://github.com/Alamar-Biosciences/NULISAseqR.

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

## Acknowledgements

We thank Alex Doe, Alex Lee, Angela Cifelli, Lucie Lee, and Yuanhui Huang for providing reagents for this study, Meirong Bai, Danielle Zhou, Roopa Comandur, and Angela Cifelli for assay development, Sydney Ho, Joseph Lau, and Tianyang Bai for antibody conjugation method development and production, April Falcone, and Khaled Alganem for assistance with figures. Many thanks to the patients and their families who volunteered to donate blood. The authors express their gratitude to the University Hospital Bonn for providing infrastructure, and transportation possibilities during the phase of sample collection. The authors thank Rolf von Uslar for interconnecting the project and the medical soldiers from the German Armed Forces for collecting samples. This project was supported in part by National Cancer Institute Small Business Innovation Research grant 1R43CA254548-01 to Y.L., the Ministry for Science and Education of the Republic of Germany COVIMMUNE/01KI20343, BMBF and the German Research Foundation (DFG) projects: Excellence Cluster EXC 2151 (ID: 390873048) and IRTG2168 (ID: 272482170) to E.L and S.V.S.

## Author contributions

Y.L. conceived the NULISA concept and led the project with X.M. Y.L., W.F., Y.Y., A.G., and S.C. designed the NULISA system. Y.L., W.F., Y.Y., and X.M. designed the study. S.V.S., E.L., X.M., and Y.L. designed the COVID-19 study. X.M. wrote the manuscript with the help of J.B., Y.L., W.F., Y.Y., A.K., Q.H., D.K., and S.V.S. Q.H., I.A., A.S., A.K., M.M., K.C., X.X., and S.I. performed the experiments. J.B., X.M., D.K., and K.B. performed the data analyses. D.K., J.B., and K.B. developed the bioinformatic pipelines. T.Y., R.N., L.W., M.Y., J.H., X.Q., and B.Z. performed assay development. K.E., J.H., L.Z., W.X., and L.F. performed reagent development and manufacturing. Y.Y., C.L., and C.H.P. developed the antibody conjugation protocol. S.C., K.Z., and C.P. built the automated assay system. S.V.S., A.O., J.S., and E.L. collected the samples and patient data for the COVID-19 study.

## Competing interests

W.F., J.B., Q.H., I.S.A., A.S., A.K., K.C., M.M., X.Q., X.X., S.I., T.Y., R.N., L.W., M.Y., K.E., L.Z., W.X., C.L., C.H.P., C.P., K.Z., A.G., J.H., K.S., D.K., L.F., B.Z., S.C., Y.Y., Y.L., and X.M. are employees and stockholders of Alamar Biosciences. Y.L., W.F., A.G., Y.Y., and S.C. have submitted a patent application to US Patent and Trademark Office pertaining to the methods and compositions of NULISA technology (provisional application No. 62/943,135). The other authors do not claim competing interest.
