## [Peer Review File · Nature Communications]

REVIEWER COMMENTS

Reviewer #1 (Remarks to the Author):

This manuscript reports on the development of a new version of proximity ligation assay named "NULISA," which employs pairs of antibodies conjugated to DNA oligonucleotides. The DNA-conjugated antibodies permit immunocomplex purification and generate reporter DNA containing target- and sample-specific barcodes. These barcodes are read out by next generation sequencing. The presented approach yields the unprecedented level of sensitivity and multiplexity. One of the innovations of this study is the approach of designing oligonucleotide DNA-antibody conjugates to use their DNA elements for both immunocomplex purification and DNA reporter generation. Another innovation is the dual capture and release mechanism that effectively enables the suppression of assay background. The study performed comprehensive benchmarking of NULISA by comparing its performance with those of other existing immunoassay methods. The study has a high impact and deserves consideration for publication. With its remarkable performance, NULISA addresses current challenges in multiplexed blood proteomics analysis, which requires high sensitivity (LOD ~ aM) and multiplexity (> 200) with a large (7-log) dynamics range. However, some of the ways in which the authors present data and make claims raise some questions. They should be addressed in a revised manuscript.

Specifically,

1. The authors claim that a 9.6-log dynamic range was demonstrated by the NULISAseq 200plex assay. However, this claim is misleading. The data in Fig. 3a clearly show that the dynamic range is dependent on each analyte species. Some analytes yield a larger dynamic range (~6 log) while others yield a smaller range (~3 log). The dynamic range of 9.6 log spanned across all the tested analytes does not validate the superiority of the assay performance.
2. Multiplexed immunoassay measurements tend to result in a large LOD as discussed in the manuscript. This may apply to NULISAseq as well. The specificity is defined as $1 - (\text{maximum non-target pool count} - \text{background}) / (\text{average target pool counts} - \text{background})$ in this study. If the authors claim the specificity of 99% for the 200-plex NULISAseq, it means the signal-to-noise ratio of the assay is SNR = 100. The specificity value should also depend on the analyte and background protein concentrations. For what analyte/background concentrations is this 99% specificity value? Is this value still attainable for a sample with a target at ~aM and background at > 1nM?
3. One of the potential weaknesses of this assay technology appears to be the need for the cumbersome, expensive, time-consuming assay process that involves reagent preparation, sample-reagent mixing/incubation, immunocomplex purification, ligated reporter oligonucleotide generation,

and reporter DNA amplification and quantification by qPCR or NGS. This issue is addressed to some degree by using a preprogrammed liquid handler. But I don't see the whole process as fully automated. How would the authors address a potentially significant barrier to broader use of the technology due to its enormous cost, resource, and time requirements?

Reviewer #2 (Remarks to the Author):

Synopsis: In this work, the authors present a NULISA, a highly sensitive and wide dynamic range immunoassay based on proximity ligation with the key addition a novel immunocomplex purification step. Immunocomplex purification is achieved by capture antibody conjugation with partially dsDNA containing a poly-A tail to enable pulldown purification by oligo-dT beads and detection antibody conjugation with partially dsDNA containing a biotin group for pulldown purification by streptavidin beads. This dual purification strategy is claimed to lower background by 10,00x and enable a >7 order of magnitude dynamic range for the assay. The manuscript demonstrates both single and highly-multiplex (200x) NULISA assays detection for an inflammation panel. The assay is assessed for dynamic range, precision, cross-reactivity, and sample detectability. Additionally, NULISA with a sequencing readout was compared to Olink and Meso Scale Diagnostics panels. In comparison, NULISA claims equal or better detectability compared to the other panels for a majority of shared targets. The authors further compared the Olink panel with NULISA for identification of differentially expressed proteins between patients with inflammatory diseases. Lastly, NULISA is used to characterize host response to SARS-CoV2 infection and demonstrates promising capability to detect multiple interferons. Overall, this work represents an advancement in highly sensitive, multiplexed detection of protein biomarkers and is worthy of consideration for publication in this journal. However, the work suffers from a number of problems, mainly arising from incomplete descriptions of the assay steps, and somewhat overblown claims that are not substantiated with data. Given this, I suggest a major revision (detailed critiques below).

Specific Comments

The claim that NULISA possesses > 7 orders of magnitude in dynamic range is not substantiated sufficiently. For example, in Figure 1b the authors do show a >7 order of magnitude dynamic range for IL4, but show an approximately 4 order of magnitude dynamic range for most targets in the panel in Figure 3a. The authors also remark in lines 158-159 "Most assays showed a dynamic range greater than three logs, covering at least two logs above and one log below the endogenous levels in healthy donors." The initial claim suggests that their assay is capable of > 7 order of magnitude dynamic range but the later results seem to contradict this. Critically, it is unclear if samples in 1b were diluted for the assay or if hot/cold antibody mixing was used to tune the assay to cover different parts of the >7 order of magnitude dynamic range. Not having 7 log dynamic range is not a problem, but making claims that are misleading / contradictory is. I strongly suggest that the authors describe the procedure for Fig 1b in detail and modify their claims of dynamic range accordingly.

The authors make strong claims of superior sensitivity, especially in comparison to their competitors (Olink) without sufficient substantiation. To convince this reviewer, a more thorough comparison of the detection of low abundance analytes is needed. For example, in line 234, the authors state that IL4, IL5, IL11, IL13, IL20, IL33, IL2RB are “low abundance” targets and NULISA measurements are more sensitive. Unfortunately, this appears to be “cherry-picking” a few analytes to their advantage. The authors should define what “low-abundance” means (less than 1 fM?) and compare all analytes that fall into that category. For example, a more complete set of plots similar to S5/S6 would be useful for all “low abundance analytes” would be useful (and not just the selected few).

The authors claim that the major break-through is the double-purification strategy that lowers the background signal by 4 logs. Since this is the main novelty of the assay, this needs to be substantiated further. Currently, the 4 log reduction (line 297) is shown for only IL4 (as shown in Fig 1b). However, this needs to be shown for many more targets to be convincing.

There is a general tendency to make claims without showing data/results. For example, where is 500-plex capability demonstrated (line 304)?? Either show the data/results or please remove extraneous claims. In another example (line 357), the authors state that switching antibody pair for CCL3 increased detectability to 100%. Although these may be true these claims are not supported by data. The authors are *strongly* advised to go through the entire manuscript and remove extraneous claims that are not supported by data.

Along similar lines, there is a general tendency to over-sell their results, which is not necessary. For example, in lines 244-246, “high concordance” might be too strong of a statement with 36 targets. There seem to be a fair amount of disagreement for the samples that were not significant. A number here would be appreciated especially since the correlation of the estimated fold-changes were $r=0.67$. Again, the authors are *strongly* advised to go through the entire manuscript and tone-down their claims to better reflect reality. The results are impressive and should speak for themselves. In fact, overselling the results cheapens the quality of the manuscript.

I assume that the same Ab-oligo conjugated batched could not have been used for PLA and NULISA by virtue of the oligos being different. Given that the background signals would depend on the purity of the Ab-oligos, the authors should show the yield and purity of all antibody—oligo conjugates used in the paper.

In fig 2, it is confusing/misleading when the data are shown on separate plots. Calibration curves for targets such as 2a could be plotted together per target (single plex v 200plex). Furthermore, the slopes (Hill coefficients) appear to be different for some targets. Why would this be the case?

Why were only LIF, IL5 and IL13 picked to show LoD in Fig 2a? To quantify as shown in Fig 2b, authors also needed standard curves for all 11 targets for the singleplex. Please show the binding curves of these other targets as well.

Why is there a difference in the definition of LOD for singleplex and multiplex NULISA? I.e. 2.5xSD vs 3xSD?

Reviewer #3 (Remarks to the Author):

This paper describes an interesting method that improves the analytic sensitivity of the PLA method, and in particular decreases the background, making the detection of proteins at very low levels possible.

In general the paper is well written and most parts of the methods described with sufficient detail. Since this method is the subject of a previous patent and the basis of commercial developments some part of the methods lack details (such as the composition of the various buffers and reagent mixes used).

There are, however, a number of points that need to be addressed by the authors:

1. There is no mentioning of the source (commercial supplier) or nature (monoclonal, polyclonal?) of the antibodies used in the assays.
2. The tests of the effect of multiplexing of the assay are adequate as a start (using a few targets at a time in different combinations), but the fact is that the full 200-plex needs to be tested in order to claim that such a high degree of multiplexing can be tolerated without a reduction of the sensitivity of any assay by cross-hybridization or increase of background.

The data shown by increasing the background of mouse EGRF for human EGRF detection is not very relevant to this point. One of the major advantages of the main competing platform, the Olink multiplex PEA, is that it appears to tolerate very high levels of multiplexing, reducing the cost per protein analysis.

3. In the abstract the NULISA is compared to the early PLA, and NULISA is stated to be 10,000-fold more sensitive. While this may be true, it is actually not very interesting because PLA is not being used for large scale protein analyses anymore. Instead it is more relevant to compare it to Olink multiplex PEA,

SomaLogic and SIMEO. In Figure 1 b and c, I would like to see the curves also for the Olink PEA as a comparison. This is highly relevant because later on in the paper a main part of the comparison between analysis platforms is between NULISA and the Olink multiplex PEA.

4. This brings me to the discussion of sensitivity (LOD). The authors claim attomolar detection, which sounds very impressive. This may be true in very specific situations, but how realistic is this in clinical samples, and how does it relate to the highly multiplex Olink PEA?

Assuming an LOD of 100 attomolar, this correspond to about 10 molecules per ul of plasma or 200 molecules for a 20 ul reaction, as described in the paper. For comparison, many of the Olink assays have an LOD of 0.02 pM or about 1000 molecules per ul plasma, or 2000 molecules in a 2 ul Olink reaction. Thus, there could be an approximate 10-fold higher sensitivity for the NUSISA. Consistent with this the NULISA did show some results for samples that were under LOD in the Olink assay. However, it remains to be shown how the two assay compare once the NULISA 200-plex assay has been characterised (in terms of cross effects on the background). If the authors include a comparison in Figure 1 as suggested above, this would provide a good basis for realistic discussions of the pros and cons of the different platforms for protein analysis.

General statement:

We are grateful to the reviewers for their careful review of our work and their constructive comments. We believe that in addressing these comments we have strengthened our work significantly. We are hopeful that the reviewers agree that the manuscript is now suitable for publication. Key additions in this revision include:

1. New data in Supplementary Fig. 1 showing two additional examples of NULISA achieving attomolar level sensitivity by background suppression
2. New Supplementary Fig. 3 showing all 11 pairs of 200-plex and single-plex standard curves to illustrate the similarly low LODs between 200-plex and single-plex
3. New data in Supplementary Fig. 7 and Supplementary Data 3 comparing the LODs and LLOQs of NULISA and Olink PEA, providing further support for NULISA's superior sensitivity
4. New data in Supplementary Fig. 10 showing how different antibodies impacted NULISA assay performance for CCL3
5. New Supplementary Data 5 listing the antibodies used in the 200-plex inflammation panel

Please find our point-by-point responses below. New or modified parts in the revised manuscript are highlighted by tracked changes. Line numbers below correspond to those in the track changed version of the manuscript.

REVIEWER COMMENTS

Reviewer #1 (Remarks to the Author):

This manuscript reports on the development of a new version of proximity ligation assay named "NULISA," which employs pairs of antibodies conjugated to DNA oligonucleotides. The DNA-conjugated antibodies permit immunocomplex purification and generate reporter DNA containing target- and sample-specific barcodes. These barcodes are read out by next generation sequencing. The presented approach yields the unprecedented level of sensitivity and multiplexity. One of the innovations of this study is the approach of designing oligonucleotide DNA-antibody conjugates to use their DNA elements for both immunocomplex purification and DNA reporter generation. Another innovation is the dual capture and release mechanism that effectively enables the suppression of assay background. The study performed comprehensive benchmarking of NULISA by comparing its performance with those of other existing immunoassay methods. The study has a high impact and deserves consideration for publication. With its remarkable performance, NULISA addresses current challenges in multiplexed blood proteomics analysis, which requires high sensitivity (LOD ~ aM) and multiplexity (> 200) with a large (7-log) dynamics range. However, some of the ways in which the authors present data and make claims raise some questions. They should be addressed in a revised manuscript.

Response: We appreciate the careful review and positive feedback on our work by this reviewer.

Specifically,

1. The authors claim that a 9.6-log dynamic range was demonstrated by the NULSAseq 200plex assay. However, this claim is misleading. The data in Fig. 3a clearly show that the dynamic range is dependent on each analyte species. Some analytes yield a larger dynamic range (~6 log) while others yield a smaller range (~3 log). The dynamic range of 9.6 log spanned across all the tested analytes does not validate the superiority of the assay performance.

Response: Thanks for pointing out this potential point of confusion, which we believe was caused by the use of two different types of dynamic ranges in the manuscript. The assay-specific dynamic range is the quantifiable range for a specific target, which is an assay performance metric. The panel-wide dynamic range (9.6-log in this case) is the accumulative dynamic range across all targets in the NULISAseq panel. It is not a measure of assay performance but a measure of the coverage of both low and high abundance proteins in the NULISAseq panel. We use this metric to highlight how NULISAseq addresses the dynamic range challenge in plasma proteomics (human plasma proteome is generally believed to span a 12-log range in concentrations) by using a hot/cold mixing signal tuning strategy to multiplex targets at vastly different concentrations in a single undiluted sample and reaction. We have clarified this in the text by referring to the 9.6-log dynamic range as accumulative dynamic range, lines 173-174, 216-217. Also note that the assay-specific dynamic ranges could be different between single-plex assays and those from the 200-plex assay, for reasons noted in lines 168-173.

2. Multiplexed immunoassay measurements tend to result in a large LOD as discussed in the manuscript. This may apply to NULSAseq as well. The specificity is defined as $1 - (\text{maximum non-target pool count} - \text{background}) / (\text{average target pool counts} - \text{background})$ in this study. If the authors claim the specificity of 99% for the 200-plex NULSAseq, it means the signal-to-noise ratio of the assay is SNR = 100. The specificity value should also depend on the analyte and background protein concentrations. For what analyte/background concentrations is this 99% specificity value? Is this value still attainable for a sample with a target at ~aM and background at > 1nM?

Response: We agree that traditional multiplexed immunoassays have been associated with larger LODs as the level of multiplexing increases. We believe this is one of the challenges NULISA technology has addressed, as demonstrated in our 200-plex vs. single-plex comparison shown in Fig. 2a and the newly added Supplementary Fig. 3 which show similar LODs between 200plex and single-plex. Regarding the 99% specificity statement, we agree with this reviewer that the signal to noise of an assay is dependent on the concentrations of both the analyte and background protein. Our experiment shown in Fig. 3c and Supplementary Fig. 5 was designed to detect cross reactivity to nontarget proteins at a fixed concentration, 20 pM for both target analyte and nontargets (this information was added in lines 195-196 of the main text and the legend for Supplementary Fig. 5). Our definition of specificity in the original manuscript was $1 -$

cross reactivity (where cross-reactivity=noise/signal). We realize this did not capture other aspects of specificity, such as the complete protein background, highly variable protein concentrations, and matrix effects. Therefore, we have modified the text to use the term “cross reactivity” to differentiate it from “specificity”, see text lines 202-205. Cross reactivity is a commonly used measure of specificity in other multiplex immunoassay platforms. In addition, we would like to point out that NULISAseq has 4 levels of specificity control built in: 1) dual antibody recognition of target protein; 2) double purification of immunocomplex prior to signal generation via dT and streptavidin beads; 3) principle of proximity ligation requiring close proximity of two antibodies to generate signal (DNA reporter); 4) discrimination of events from binding by cognate pairs of antibodies from those from noncognate pairs of antibodies resulting from either unspecific target binding or heterophilic antibody-antibody interactions by matching predesignated target-specific barcodes (TMIs) by NGS. To our knowledge, having these four levels of built-in specificity in an immunoassay is unprecedented, and it explains why NULISA assay not only has high specificity (low noise) but also best-in-class sensitivity (high signal/noise ratio).

3. One of the potential weaknesses of this assay technology appears to be the need for the cumbersome, expensive, time-consuming assay process that involves reagent preparation, sample-reagent mixing/incubation, immunocomplex purification, ligated reporter oligonucleotide generation, and reporter DNA amplification and quantification by qPCR or NGS. This issue is addressed to some degree by using a preprogrammed liquid handler. But I don't see the whole process as fully automated. How would the authors address a potentially significant barrier to broader use of the technology due to its enormous cost, resource, and time requirements?

Response: we agree that NULISA technology consists of a few more steps than a traditional immunoassay, and it would be more difficult to adopt without automation. Indeed, full automation was the goal at the start of R&D. As stated in the manuscript, the magnetic beads-based capture and purification protocol is very amenable to automation. In the final protocol, there are no sample dilution steps and no manual reagent preparation steps. The complexity of reagent preparation, including antibody conjugation, purification, and signal tuning is handled at manufacturing, where we have automated processes to ensure consistency. Our ARGO™ instrument performs all steps from sample/reagent mixing, immunocomplex purification, reporter generation to qPCR quantification for single-plex assays, or to an NGS-ready DNA library for multiplexing. NGS is performed separately on an Illumina sequencer with minimal hands-on time. The level of automation of NULISA with < 30 mins hands-on time and the within-one-day total time from sample to results is as important as the improved performance for broad adoption. In addition, the cost of the instrument and reagents will be comparable to existing platforms on the market on a per sample and per data point basis. For example, compared to Olink PEA, NULISA uses two additional components– the paramagnetic oligo-dT beads and streptavidin beads, both of which are low-cost items so from cost point of view it will not be a barrier for NULISA adoption.

Reviewer #2 (Remarks to the Author):

Synopsis: In this work, the authors present a NULISA, a highly sensitive and wide dynamic range immunoassay based on proximity ligation with the key addition a novel immunocomplex purification step. Immunocomplex purification is achieved by capture antibody conjugation with partially dsDNA containing a poly-A tail to enable pulldown purification by oligo-dT beads and detection antibody conjugation with partially dsDNA containing a biotin group for pulldown purification by streptavidin beads. This dual purification strategy is claimed to lower background by 10,00x and enable a >7 order of magnitude dynamic range for the assay. The manuscript demonstrates both single and highly-multiplex (200x) NULISA assays detection for an inflammation panel. The assay is assessed for dynamic range, precision, cross-reactivity, and sample detectability. Additionally, NULISA with a sequencing readout was compared to Olink and Meso Scale Diagnostics panels. In comparison, NULISA claims equal or better detectability compared to the other panels for a majority of shared targets. The authors further compared the Olink panel with NULISA for identification of differentially expressed proteins between patients with inflammatory diseases. Lastly, NULISA is used to characterize host response to SARS-CoV2 infection and demonstrates promising capability to detect multiple interferons. Overall, this work represents an advancement in highly sensitive, multiplexed detection of protein biomarkers and is worthy of consideration for publication in this journal. However, the work suffers from a number of problems, mainly arising from incomplete descriptions of the assay steps, and somewhat overblown claims that are not substantiated with data. Given this, I suggest a major revision (detailed critiques below).

Response: We appreciate the overall positive assessment of our work and helpful constructive critiques by this reviewer. We have added additional supporting data in Supplemental Information and toned down our interpretation statements throughout in this revision.

Specific Comments

The claim that NULISA possesses > 7 orders of magnitude in dynamic range is not substantiated sufficiently. For example, in Figure 1b the authors do show a >7 order of magnitude dynamic range for IL4, but show an approximately 4 order of magnitude dynamic range for most targets in the panel in Figure 3a. The authors also remark in lines 158-159 “Most assays showed a dynamic range greater than three logs, covering at least two logs above and one log below the endogenous levels in healthy donors.” The initial claim suggests that their assay is capable of > 7 order of magnitude dynamic range but the later results seem to contradict this. Critically, it is unclear if samples in 1b were diluted for the assay or if hot/cold antibody mixing was used to tune the assay to cover different parts of the >7 order of magnitude dynamic range. Not having 7 log dynamic range is not a problem, but making claims that are misleading / contradictory is. I strongly suggest that the authors describe the procedure for Fig 1b in detail and modify their claims of dynamic range accordingly.

Response: Thanks for pointing out these apparent inconsistencies. Fig. 1b and 1c were from single-plex assays using qPCR readout. No sample dilution or hot/cold antibody mixing was performed so these single-plex assays were configured for maximal sensitivity and dynamic range. However, the dynamic ranges shown in Fig. 3a were from the 200-plex with NGS readout. The main goal of the multiplex assay was to be able to quantitate both low and high abundance targets in a single undiluted sample and maintain the highest sensitivity for low abundance targets. Maximal dynamic range is not possible due to limited capacity of NGS sequencing reads and was not the goal as long as sample measurements are within the detectable range. As presented in the manuscript, to achieve this, we devised the hot/cold antibody mixing strategy to intentionally lower the sensitivity for high abundance targets so that they don't consume bulk of the sequencing reads. Therefore, the lower dynamic range values in Fig. 3a were partly due to this intentional decrease in sensitivity for high abundance targets. The other reason for the limited dynamic range is the limited capacity of total sequencing reads in an NGS readout (~500M reads in this case). The range of the calibrator standard concentrations for each assay was truncated at the high end, i.e., it did not reach the hook region of a full standard curve. This was necessary because we didn't want the high concentration points to consume most of the sequencing capacity and negatively impact the low end of the dilution curve. As a result, the upper limit of quantification (ULOQ) values were simply the highest concentrations (if meeting our <30% CV and <30% recovery error criteria) we used in this experiment, not the true ULOQ, for most of these assays. Hence, the dynamic ranges for each of the individual assays were 3-4 logs in NULISAseq. We have added more detail to Fig. 1b in the legend and added a new section to explain the difference between single-plex and 200-plex in terms of dynamic range (lines 168-175).

The authors make strong claims of superior sensitivity, especially in comparison to their competitors (Olink) without sufficient substantiation. To convince this reviewer, a more thorough comparison of the detection of low abundance analytes is needed. For example, in line 234, the authors state that IL4, IL5, IL11, IL13, IL20, IL33, IL2RB are "low abundance" targets and NULISA measurements are more sensitive. Unfortunately, this appears to be "cherry-picking" a few analytes to their advantage. The authors should define what "low-abundance" means (less than 1 fM?) and compare all analytes that fall into that category. For example, a more complete set of plots similar to S5/S6 would be useful for all "low abundance analytes" would be useful (and not just the selected few).

Response: This is a great comment about how to define low abundance. There is no agreed to definition or cutoff, so we operationally classify proteins at low pg/mL or lower as low abundance, roughly corresponding to the limit of detection of current ELISA assays. However, it is not possible to classify all 200 proteins in the 200-plex assay as low or high abundance because for most of these proteins their absolute concentrations in the blood are not well-established. Therefore, to address the concern of "cherry picking" proteins in the discussion, we have deleted this discussion in this revision, see line 250-254.

To follow this reviewer's suggestion to make a more thorough comparison with Olink on low abundance protein detection, in the revised manuscript, we added a new analysis comparing LOD/LLOQ and detectability for proteins that require maximal sensitivity on each platform, defined as those that do not use hot/cold antibody mixing-based signal tuning in NULISAseq and use undiluted samples, i.e., belonging to the lowest abundance block, in the Olink Explore assay. The results are shown in Supplementary Fig. 7 and Supplementary Data 3, and presented in a new section in the main text, lines 256-279 and in Methods, lines 616-626. This new analysis demonstrated that for the 74 proteins shared between NULISAseq 200-plex and Olink Explore 3072, NULISA showed 250-fold and 65-fold lower median LODs and LLOQs than Olink, respectively. For the subset of 45 proteins from these 74 for which detectability data from both platforms were available, the lower LODs of NULISAseq translated into higher detectability in plasma samples by NULISAseq (95% vs. 83%, $p=0.016$), e.g., IL4, IL5, IL13, IL20 and IL33 showing the largest differences in LOD between NULISA and Olink also had the largest differences in detectability between the two platforms. Furthermore, 4 of these 5 proteins, IL4, IL5, IL20 and IL33, also were identified to be significantly associated with inflammatory diseases (Fig. 4d-e) by NULISAseq but not by Olink. These three lines of evidence together support our conclusion that NULISAseq has superior sensitivity to Olink, at least for the low abundance proteins as defined here.

The authors claim that the major break-through is the double-purification strategy that lowers the background signal by 4 logs. Since this is the main novelty of the assay, this needs to be substantiated further. Currently, the 4 log reduction (line 297) is shown for only IL4 (as shown in Fig 1b). However, this needs to be shown for many more targets to be convincing.

Response: Points well taken. Please see the new Supplementary Fig. 1 showing two additional examples where NULISA lowered background by 30,000-fold (IL6) and 3,600-fold (CXCL5). While we have done only limited head-to-head comparisons with PLA, we believe that the benefit of the double-purification strategy should be broad, even if the exact extent of benefit may be different for different targets or antibodies. This is supported by the superior LOD/LLOQ of NULISAseq for low abundance proteins compared to Olink in the analysis discussed above, considering that NULISA and PEA share three specificity control mechanisms, e.g., dual antibody recognition, proximity mediated signal generation and barcode matching via NGS, except for the lack of immunocomplex purification in PEA.

There is a general tendency to make claims without showing data/results. For example, where is 500-plex capability demonstrated (line 304)?? Either show the data/results or please remove extraneous claims. In another example (line 357), the authors state that switching antibody pair for CCL3 increased detectability to 100%. Although these may be true these claims are not supported by data. The authors are *strongly* advised to go through the entire manuscript and remove extraneous claims that are not supported by data.

Response: We have removed the reference to the 500-plex feasibility data because it is a complex dataset to include and does not add significantly to this study; see main text line 349.

We have added a new Supplementary Fig. 10 to show the improved assay performance for CCL3 after switching to a new antibody pair. We have gone through the manuscript to ensure there are no unsupported claims.

Along similar lines, there is a general tendency to over-sell their results, which is not necessary. For example, in lines 244-246, “high concordance” might be too strong of a statement with 36 targets. There seem to be a fair amount of disagreement for the samples that were not significant. A number here would be appreciated especially since the correlation of the estimated fold-changes were $r=0.67$. Again, the authors are *strongly* advised to go through the entire manuscript and tone-down their claims to better reflect reality. The results are impressive and should speak for themselves. In fact, overselling the results cheapens the quality of the manuscript.

Response: Points well taken. We should have been more careful with these statements. We have deleted the “high concordance” statement, main text line 289-290, and modified the claim about the fold-change correlation from “good” to significant, main text line 291. We have gone through the entire manuscript text to tone down similar claims, e.g., deleting “greatly” in line 346, changing “major” to “significant” in line 410, etc.

I assume that the same Ab-oligo conjugated batched could not have been used for PLA and NULISA by virtue of the oligos being different. Given that the background signals would depend on the purity of the Ab-oligos, the authors should show the yield and purity of all antibody—oligo conjugates used in the paper.

Response: The same Ab-oligo conjugates were used for both PLA and NULISA for Fig. 1b and the new Supplementary Fig. 1. Since PLA and NULISA both use proximity ligation for signal generation, the same oligo design can be used for both assays. By using the same Ab-oligo conjugates and same assay buffer in the comparison experiment, we could make a direct comparison between the two assay protocols. We agree that antibody-DNA purity is critical to signal and background. We have implemented automation for the conjugation and purification processes including a run control using IL4 for each batch. We run the final conjugated materials on protein gels to determine purity (>90%) and successful conjugation. We have added this detail in Methods lines 443-445.

In fig 2, it is confusing/misleading when the data are shown on separate plots. Calibration curves for targets such as 2a could be plotted together per target (single plex v 200plex). Furthermore, the slopes (Hill coefficients) appear to be different for some targets. Why would this be the case?

Response: We did not plot both curves in one plot because the qPCR data and the NGS data underwent different transformations and are on different scales. In order to plot the qPCR C_q data such that increasing concentration leads to increasing signal (the convention for immunoassay data), a transformation of $2^{(37-C_q)}$ was applied, where the 37 is an arbitrary

number that shifts the signal values. Additionally, in order to obtain positive, normally distributed values, the NULISAseq 200-plex normalization procedure involves rescaling the data by an arbitrary constant (10^4), see Methods lines 542-553. For these reasons, while the idea of plotting both qPCR and NGS data on the same plots is appealing, it is not very meaningful as the horizontal asymptotes of the signal curves cannot be directly compared. However, the back-transformed LOD values derived from the qPCR and NGS curves can be directly compared. Regarding the slopes of the 11 pairs of standard curves (Supplementary Fig. 3), the actual slopes (see Table below) are actually more similar between single-plex and 200-plex (single-plex slope mean=0.99, 200-plex slope mean=0.98, paired t test $p=0.25$) than they appear visually in Supplementary Fig. 3:

Target	Single-plex	200-plex
CXCL8	1.03E+00	1.02E+00
IFNA1	9.78E-01	9.72E-01
IFNL1	9.64E-01	9.69E-01
IL4	9.91E-01	9.65E-01
IL5	1.01E+00	9.69E-01
IL6	9.94E-01	9.99E-01
IL10	1.04E+00	9.59E-01
IL13	9.95E-01	9.76E-01
IL20	9.59E-01	9.91E-01
LIF	9.80E-01	9.64E-01
TSLP	9.47E-01	9.77E-01

Why were only LIF, IL5 and IL13 picked to show LoD in Fig 2a? To quantify as shown in Fig 2b, authors also needed standard curves for all 11 targets for the singleplex. Please show the binding curves of these other targets as well.

Response: Please find the requested 11 pairs of plots in the new Supplementary Fig. 3. The LODs were similar in magnitude between single-plex and 200-plex for the 11 targets.

Why is there a difference in the definition of LOD for singleplex and multiplex NULISA? I.e 2.5xSD vs 3xSD?

Response: This is a good question. For this study, LOD definitions were chosen to facilitate direct comparison with other quantification methods. In the case of single-plex NULISA, we wanted to compare it to Quanterix Simoa assays, so we chose their definition of LOD (2.5xSD). In the case of NULISAseq multiplex, we chose the definition of 3xSD used by MSD and Olink to facilitate comparison of LODs. To make this clearer, we now state this in the manuscript, see lines 558- 563.

Reviewer #3 (Remarks to the Author):

This paper describes an interesting method that improves the analytic sensitivity of the PLA method, and in particular decreases the background, making the detection of proteins at very low levels possible.

In general the paper is well written and most parts of the methods described with sufficient detail. Since this methods is the subject of a previous patent and the basis of commercial developments some part of the methods lack details (such as the composition of the various buffers and reagent mixes used).

Response: We appreciate the positive feedback on our work and understanding of the need to protect confidential formulation information by this reviewer.

There are, however, a number of points that needs to be adressed by the authors:

1. There is no mentioning of the source (commercial supplier) or nature (monoclonal, polyclonal?) of the antibodies used in the assays.

Response: We have added a new Supplementary Data 5 file listing the antibodies used in this study and referred to it in Methods lines 443-445. We included Alamar's unique IDs, host species, and clonality (monoclonal or polyclonal). This should provide the reader with an understanding of the nature of the antibodies used in our assays and the ability to uniquely identify specific antibodies in case someone wishes to replicate our results. However, the exact commercial sources are Alamar's proprietary and confidential information.

2. The tests of the effect of multiplexing of the assay are adequate as a start (using a few targets at a time in different combinations), but the fact is that the full 200-plex needs to be tested in order to claim that such a high degree of multiplexing can be tolerated without a reduction of the sensitivity of any assay by cross-hybridization or increase of background.

Response: We agree our comparison between 200-plex and single-plex was limited as we focused our comparison on low abundance targets that require ultra-high sensitivity. However, testing the impact of multiplexing on every target in the full 200-plex will be too time consuming and costly to be practical. We have modified our statement to reflect our limited testing in the main text, lines 133-134.

The data shown by increasing the background of mouse EGRF for human EGRF detection is not very relevant to this point. One of the major advantages of the main competing platform, the Olink multiplex PEA, is that it appears to tolerate very high levels of multiplexing, reducing the cost per protein analysis.

Response: We agree the mouse EGFR experiment using a single-plex NULISA assay was not directly relevant to background issues caused by multiplexing. However, the cross-reactivity

experiment presented in Supplemental Fig. 5 was relevant to this point and this is the way other multiplex assays are evaluated as well. The full 200-plex antibody reagents were used to analyze all recombinant protein pools, and we assessed cross reactivity of each antibody pair to all antigens. The results demonstrated <1% cross reactivity for >90% of the assays.

Conceptually, NULISA includes all the elements of specificity mechanisms in Olink PEA and the additional double capture/purification mechanisms, which should support high multiplexing levels as well as if not better than Olink does. Furthermore, NULISA makes multiplexing even simpler than Olink PEA due to 1) the use of PLA as opposed to PEA (discussed in lines 346-355) and 2) the use of hot/cold antibody mixing which enables analysis of low and high abundance targets in the same reaction/sample, whereas in Olink PEA, targets are split into multiple subsets for analysis in separate reactions in different “abundance “blocks” created by sample dilutions.

3. In the abstract the NULISA is compared to the early PLA, and NULISA is stated to be 10,000-fold more sensitive. While this may be true, it is actually not very interesting because PLA is not being used for large scale protein analyses anymore. Instead it is more relevant to compare it to Olink multiplex PEA, SomaLogic and SIMEO. In Figure 1 b and c, I would like to see the curves also for the Olink PEA as a comparison. This is highly relevant because later on in the paper a main part of the comparison between analysis platforms is between NULISA and the Olink multiplex PEA.

Response: We agree the 10,000-fold improvement in LOD of single-plex NULISA compared to PLA is not relevant to the comparison with Olink PEA, as the purpose of this experiment was to highlight the NULISA performance improvement over PLA. It's not possible for us to conduct a direct comparison of single-plex NULISA vs single-plex Olink PEA for two reasons: 1) Olink doesn't provide single-plex assays; 2) it will not be the same antibody pair on both platforms for an apples-to-apples comparison. To address the reviewer's question, we have added a new analysis comparing LODs and LLOQs of NULISAseq and Olink PEA shown in Supplementary Fig. 7 and Supplementary Data 3 and discussed in main text lines 256-279. In this analysis, we used the 74 shared low abundance targets between NULISAseq 200-plex and Olink Explore 3072 that were not tuned by hot/cold mixing in NULISA or by sample dilution in Olink. The results showed that the median LOD and LLOQ of NULISAseq was 250-fold and 65-fold lower than those of PEA, respectively.

4. This brings me to the discussion of sensitivity (LOD). The authors claim attomolar detection, which sounds very impressive. This may be true in very specific situations, but how realistic is this in clinical samples, and how does it relate to the highly multiplex Olink PEA? Assuming an LOD of 100 attomolar, this correspond to about 10 molecules per ul of plasma or 200 molecules for a 20 ul reaction, as described in the paper. For comparison, many of the Olink assays have an LOD of 0.02 pM or about 1000 molecules per ul plasma, or 2000 molecules in a 2 ul Olink reaction. Thus, there could be an approximate 10-fold higher sensitivity for the NUSISA. Consistent with this the NULISA did show some results for samples that were under

LOD in the Olink assay. However, it remains to be shown how the two assay compare once the NULISA 200-plex assay has been characterised (in terms of cross effects on the background). If the authors include a comparison in Figure 1 as suggested above, this would provide a good basis for realistic discussions of the pros and cons of the different platforms for protein analysis.

Response: As explained above, we have added a new data analysis to compare the LODs of NULISAseq and Olink PEA; see main text lines 256-279. This analysis demonstrated a median 250-fold lower LOD for NULISAseq, a much larger difference than the 10-fold difference in input molecules. Another important metric for assay sensitivity in real-world clinical samples (plasma, serum, etc.) is sample detectability, the percentage of samples a target signal is detected above the target-specific LOD. This metric incorporates the assay background in its calculation, including potential cross reaction background. As shown in Supplementary Table 3, in a set of 151 plasma samples including 79 healthy controls and 72 patients with various diseases, NULISAseq demonstrated 95.9% mean detectability compared to 91.6% by Olink PEA across all 92 shared targets between the two platforms. When we focus on the 45 shared low abundance targets that were not tuned by hot/cold mixing in NULISA or by sample dilution in Olink, the mean detectability was 95% for NULISAseq and 83% for Olink PEA (Supplementary Data 3) (paired t-test $p = 0.016$). These detectability differences translated into differences in detecting differentially abundant proteins associated with inflammatory diseases, see Fig. 4c-e. These data collectively demonstrated NULISAseq's superior analytical sensitivity, higher detectability in real clinical samples, which translated into the ability to detect disease-associated changes in low abundance proteins, i.e., potential new biological insights.

REVIEWERS' COMMENTS

Reviewer #1 (Remarks to the Author):

The new proximity ligation assay "NULISA" employs pairs of antibodies conjugated to DNA oligonucleotides. The DNA-conjugated antibodies permit immunocomplex purification and generate reporter DNA containing target- and sample-specific barcodes. These barcodes are read out by next-generation sequencing. The presented approach yields an unprecedented level of sensitivity and multiplexity. The revised manuscript addresses all of the reviewer's concerns and questions and is suited for publication in Nature Comm.

A minor point: The two plots in Supplementary Fig. 1 miss the unit in their horizontal axes.

Reviewer #2 (Remarks to the Author):

In this revised version, the authors have effectively addressed most of my concerns providing satisfactory explanations and additional data. I commend the authors for their efforts. I have a couple of suggestions to further improve the manuscript (below). Once the authors implement them, I am satisfied with the revision and suggest acceptance of the paper.

1. There are two types of dynamic ranges – one for the entire assay and another for each analyte. The dynamic range for the assay is now clear. However, the description of the dynamic range for each target is still confusing. While authors confirmed a 7-log dynamic range for IL4 and p24, this range cannot be assumed to apply to all other targets, as indicated in lines 93-95. To prevent any misleading impressions, I suggest rephrasing this section to clarify that the 7-log dynamic range pertains specifically to IL4 and p24, and making it clear that other targets have narrower dynamic ranges.

2. The labeling of "Signal" on the y-axes of graphs in Fig. 2a and Supplementary Fig. S3 should be improved for clarity and consistency. Here are some suggestions to improve the expression:

a. Consider changing the label to "Normalized Signal" to reflect the nature of the values

b. Provide a clear explanation in the main text regarding the method used to calculate "signal" values for both single-plex and 200-plex measurements. This will ensure transparency and help readers better interpret the data.

Reviewer #3 (Remarks to the Author):

I find that the authors have addressed my comments in a reasonably satisfactory way, and that the revised version is overall better.

REVIEWERS' COMMENTS

Reviewer #1 (Remarks to the Author):

The new proximity ligation assay "NULISA" employs pairs of antibodies conjugated to DNA oligonucleotides. The DNA-conjugated antibodies permit immunocomplex purification and generate reporter DNA containing target- and sample-specific barcodes. These barcodes are read out by next-generation sequencing. The presented approach yields an unprecedented level of sensitivity and multiplexity. The revised manuscript addresses all of the reviewer's concerns and questions and is suited for publication in Nature Comm.

A minor point: The two plots in Supplementary Fig. 1 miss the unit in their horizontal axes.

Response: We thank the reviewer for catching this. This is fixed.

Reviewer #2 (Remarks to the Author):

In this revised version, the authors have effectively addressed most of my concerns providing satisfactory explanations and additional data. I commend the authors for their efforts. I have a couple of suggestions to further improve the manuscript (below). Once the authors implement them, I am satisfied with the revision and suggest acceptance of the paper.

1. There are two types of dynamic ranges – one for the entire assay and another for each analyte. The dynamic range for the assay is now clear. However, the description of the dynamic range for each target is still confusing. While authors confirmed a 7-log dynamic range for IL4 and p24, this range cannot be assumed to apply to all other targets, as indicated in lines 93-95. To prevent any misleading impressions, I suggest rephrasing this section to clarify that the 7-log dynamic range pertains specifically to IL4 and p24, and making it clear that other targets have narrower dynamic ranges.

Response: Yes, the dynamic range of an assay will vary depending on the target, antibodies used and the detection method. To prevent any unintended generalization of the 7-log dynamic range shown for IL4 and p24, we added the following clarification in lines 116-120:

“It should be noted that the large dynamic range shown for IL4 and p24 in these experiments was attributable to background suppression and the qPCR readout, but the degree of improvement for other targets will vary depending on the specific target and antibodies used as well as the readout method.”

2. The labeling of "Signal" on the y-axes of graphs in Fig. 2a and Supplementary Fig. S3 should be improved for clarity and consistency. Here are some suggestions to improve the expression:

- Consider changing the label to "Normalized Signal" to reflect the nature of the values
- Provide a clear explanation in the main text regarding the method used to calculate "signal" values for both single-plex and 200-plex measurements. This will ensure transparency and help readers better interpret the data.

Response: thank the reviewer for the suggestion. We have changed the Y-axis label in Fig. 2a and Supplementary Fig. 3 as suggested. We have also clarified our explanation on the calculation of normalized signal in the Methods section, lines 539- 561.

Reviewer #3 (Remarks to the Author):

I find that the authors have addressed my comments in a reasonably satisfactory way, and that the revised version is overall better.